# Bayesian Evidence-Driven Prototype Evolution for Federated Domain Adaptation

**Xiaoyang Yi**[1,2,3]**, Li Peng**[1,2,3]**, Yuru Bao**[1,2,3]**, Jian Zhang**[1,2,3] *

1 College of Cryptology and Cyber Science, Nankai University
2 Tianjin Key Laboratory of Network and Data Security Technology
3 Key Laboratory of Data and Intelligent System Security, Ministry of Education
`{xiaoyangyi,pengli,baoyuru}@mail.nankai.edu.cn,`
`zhang.jian@nankai.edu.cn`

## Abstract

Federated learning (FL), as a privacy-preserving distributed machine learning paradigm, enables clients to collaboratively train a global model without sharing local data. However, in real-world scenarios, domain shift caused by different source clients leads to structural discrepancies in the feature space, resulting in performance degradation of the global model. Although existing prototype-based FL methods offer improvements in cross-domain feature alignment, they still struggle to adapt to dynamic semantic structures and fail to continuously respond to the changing semantic separability and variance structure during training. To address this, we propose FedPTE, an FL framework with prototype topology evolution. Specifically, FedPTE treats prototype clusters as variable topological units, employing Bayesian Gaussian Mixture Models and marginal likelihood ratios on the server to perform probabilistic inference, which enables adaptive structural adjustments. Meanwhile, FedPTE introduces a stability constraint mechanism to balance the adaptability of topological evolution and training stability. By conducting prototype topology-aware contrastive learning on clients, it enhances the discriminability and cross-domain consistency of features. Experimental results demonstrate that FedPTE achieves superior performance across multiple cross-domain datasets, showcasing its strong expressiveness and generalization capability in heterogeneous domains.

## 1 Introduction

Federated learning (FL), as a privacy-preserving distributed machine learning paradigm, enables multiple clients to collaboratively train a global model without sharing local data, demonstrating significant potential across various application fields (McMahan et al., 2017; Ghari & Shen, 2024; Liu et al., 2025). However, conventional FL methods typically assume that data across clients is independently and identically distributed (IID), an assumption that often fails to hold in real-world scenarios (Gao et al., 2024; Zhang et al., 2025b; Gao et al., 2025). Particularly in healthcare, different hospitals may use varying imaging equipment, follow divergent diagnostic protocols, or serve geographically distinct patient populations, leading to substantial differences in feature distributions (Jia et al., 2024b; Chen et al., 2024a), which is a phenomenon known as domain shift (Le et al., 2024; Jiang et al., 2024). This shift is profoundly reflected in the structural discrepancies within the feature space, making it difficult for models trained on a single data source to generalize to other clients. Consequently, the global model struggles to achieve optimal performance across all clients (Jia et al., 2024a; Feng et al., 2025).

To mitigate domain shift, recent studies have introduced prototype-based FL methods (Wan et al., 2024; Fu et al., 2025a;b) to enhance cross-domain feature alignment. Some methods construct local and global prototypes simply by averaging features to align representations, which often leads to information loss and inadequate representation of challenging domains (Zhang et al., 2025a; Fu et al., 2025b). Other methods cluster features on clients and upload more cluster centers to the server

---

*Corresponding author

for prototype clustering (Huang et al., 2023; Wang et al., 2024). However, as clients participate more extensively in the FL process, the model progressively refines its estimation of feature distributions, resulting in noticeable in-domain variations within the same class. Relying on static clustering and simple averaging prevents prototypes from accurately characterizing the evolution of features, making it difficult to continuously adapt to dynamic changes in semantic separability and variance structure during training (Wu et al., 2024; Qu et al., 2025).

A more desirable option is to view the server as a center that maintains a dynamic prototype topology (Liang et al., 2024; Li et al., 2025), using criteria to determine when to split a prototype cluster to capture in-class structures or when to merge similar prototype clusters to eliminate redundancy and noise based on statistical evidence (Huang et al., 2025). From this perspective, the server treats the number and structure of prototype clusters as an evidence-driven model state. When confronted with complex distributions within a class or cross-domain neighbor conflicts, it automatically decides whether to refine certain prototype clusters to retain important information or merge approximate prototype clusters to denoise and compress representations. However, if the server drastically alters the global prototype structure in a single round, which may severely mismatch the current client representations, leading to convergence degradation. Overly conservative criteria may delay necessary refinements, while overly aggressive ones may frequently cause instability. Therefore, there is an urgent need for a prototype learning framework capable of dynamically perceiving class granularity to achieve finer-grained modeling of cross-domain feature distributions.

Armed with these insights, we propose a **Fed**erated learning framework with **P**rototype **T**opology **E**volution, FedPTE, which treats prototype clusters as variable topological units for probabilistic inference and topological maintenance to mitigate the heterogeneity of feature distributions across domains. Specifically, FedPTE employs Bayesian Gaussian Mixture Models (BGMM) and marginal likelihood ratios under a Normal-Inverse-Wishart (NIW) prior (Diebolt & Robert, 1994; Zhao et al., 2023) as statistical evidence on the server to determine whether to split a global prototype cluster into finer sub-prototype clusters or merge several approximate prototype clusters into a more robust representation. This allows the global number and topology of prototype clusters to be adaptively adjusted. To prevent training instability caused by abrupt topological changes on the server, FedPTE introduces split and merge penalty terms to restrict drastic structural changes within a single round, thereby balancing adaptability and training stability. Finally, FedPTE performs prototype topology-aware contrastive learning on clients to effectively capture in-domain variance information and achieve cross-domain feature alignment, enhancing the model's generalization capability in heterogeneous domains.

Our contributions are as follows:

- We propose a prototype topology-based FL framework, FedPTE, that treats prototype clusters as evidence-driven topological units, performing probabilistic inference and topological maintenance to alleviate feature distribution heterogeneity across domains.

- We elaborately design splitting and merging criteria based on a probabilistic model with NIW prior and marginal likelihood ratios to refine or aggregate prototype clusters, reducing reliance on empirical thresholds and static clustering.

- We incorporate stability constraints on the server to limit abrupt topological changes, and conduct prototype topology-aware contrastive learning on clients, ensuring a balance between adaptive topological adjustment and training convergence.

- Experimental results demonstrate that FedPTE achieves superior performance across multiple cross-domain datasets under various settings, indicating its expressiveness and generalization capability in heterogeneous domains.

## 2 RELATED WORK

FL with domain shift refers to the issue of feature distribution heterogeneity in FL caused by data from different domains across clients. For addressing this, FPL (Huang et al., 2023) generates unbiased prototypes and combines consistency regularization to align local features with unbiased prototypes, thereby enhancing the model's generalization capability under domain shift. FedHEAL (Chen et al., 2024b) employs a fairness aggregation strategy based on distance variance to reduce

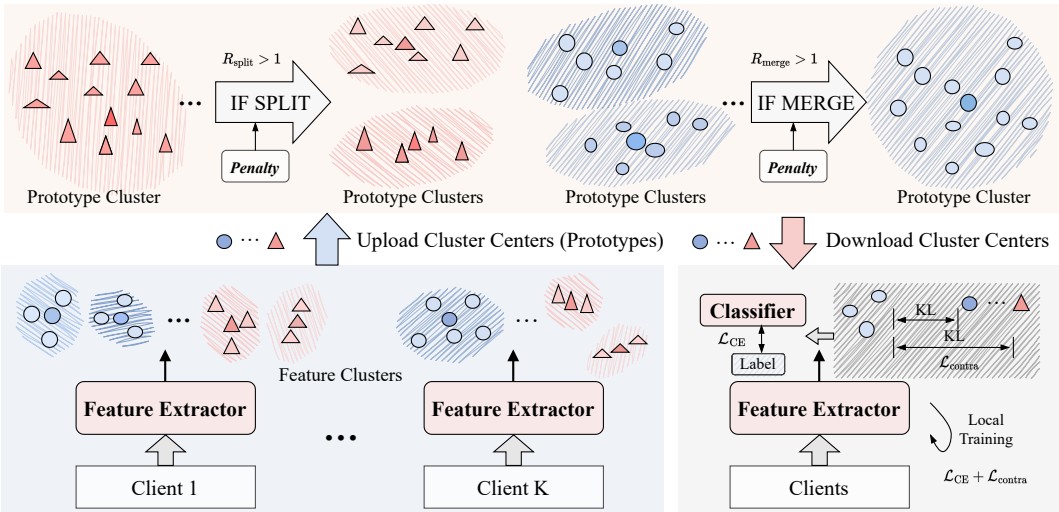

Figure 1: The framework of FedPTE. Clients first upload local prototypes to the server. The server then adapts the global prototype topology through an evidence-based decision-making process that utilizes a NIW prior and marginal likelihood ratios, regulated by penalty constraints. Finally, clients download the evolved cluster centers to guide their local training via a joint loss function.

the model's bias toward certain domains, thus improving the fairness of FL in the context of domain shift. AlignFed (Zhu et al., 2024) uses personalized feature extractors at each client to align features from different domains to a set of predefined global target points, thereby reducing feature shift among clients. FedPLVM (Wang et al., 2024) utilizes a dual-layer prototype clustering mechanism to capture variance information and introduces an $\alpha$-sparse prototype loss function to balance in-class distances, thereby improving cross-domain model performance. FedLSA (Fu et al., 2025a) leverages a global semantic-aware classifier to enhance in-class separation and designs a contrastive loss based on the von Mises-Fisher distribution to promote compactness and uniform distribution of in-class representations. FDSE (Wang et al., 2025) decomposes each layer of the neural network into a domain-agnostic feature extractor and a domain-specific shift eraser, iteratively extracting and eliminating domain shifts in features. MPFT (Zhang et al., 2025a) generates domain-specific prototypes representing local data distributions on clients and uses these prototypes to centrally train a global adapter on the server to mitigate feature distribution heterogeneity.

## 3 METHOD

**Preliminaries.** In FL, each client $k \in K$ holds its private dataset $\mathcal{D}_k = \{(\mathbf{x}_i, y_i)\}_{i=1}^{N_k}$, where $\mathbf{x}_i$ is an input sample and $y_i \in \{1, ..., m, ..., M\}$ is the corresponding class label. In FL with domain shift, different clients may belong to different domains, meaning there exists heterogeneity in the feature distributions of the same class of data across clients. Additionally, each client $k$ contains a local model $f(\cdot; \mathbf{w})$, which consists of a feature extractor $h(\cdot; \mathbf{w}_h)$ and a classifier $g(\cdot; \mathbf{w}_g)$, i.e., $\mathbf{w} = \{\mathbf{w}_h, \mathbf{w}_g\}$. The local optimization objective of the model on client $k$ is:

$$\mathcal{L}_k(\mathbf{w}) = \frac{1}{N_k} \sum_{(\mathbf{x}_i, y_i) \in \mathcal{D}_k} \ell(f(\mathbf{x}_i; \mathbf{w}), y_i) \tag{1}$$

where $\ell(\cdot, \cdot)$ is the loss function.

Prototype-based FL primarily focuses on the representations in the feature space. For a sample $\mathbf{x}_i$, its feature is $\mathbf{z}_i = h(\mathbf{x}_i; \mathbf{w}_h) \in \mathbb{R}^d$. The prototypes $\mathcal{P}_k^m = \{\mathbf{p}_k^{m,c}\}_{i=1}^{|C_{\text{local}}^m|}$ in clustering methods are obtained by clustering the features and then calculating the mean of the cluster features:

$$\mathbf{p}_k^{m,c} = \frac{1}{|\mathcal{D}_k^{m,c}|} \sum_{(\mathbf{x}_i, y_i) \in \mathcal{D}_k^{m,c}} h(\mathbf{x}_i; \mathbf{w}_h) \tag{2}$$

where $\mathcal{D}_k^{m,c}$ is the subset of cluster samples through clustering algorithm such as FINCH (Sarfraz et al., 2019), $C_{\text{local}}^m$ is the number of clusters of class $m$ on client $k$. These prototypes can be uploaded from clients to the server during communication. After processing, the server sends them back to clients to mitigate the heterogeneity caused by domain shift.

**Overview.** The framework of FedPTE is illustrated in Figure 1. It treats the global prototype set in FL as a dynamically evolvable topological structure. The server adapts the global prototype topology through an evidence-based decision-making process that utilizes a NIW prior and marginal likelihood ratios, regulated by penalty constraints, thereby more precisely characterizing the cross-domain feature distributions of classes. Meanwhile, clients leverage the evolved global cluster centers to constrain local training through a joint loss function, enhancing the discriminability and domain invariance of feature representations.

### 3.1 BAYESIAN EVIDENCE FOR TOPOLOGY ADAPTATION

In each communication round, for each class $m$, clients upload their prototypes $\mathcal{P}_k^m$. The server collects all candidate points related to class $m$ uploaded by all clients to form a prototype cluster, thereby maintaining the current global prototype topology $\mathcal{G} = \{\mathbf{g}_j\}_{j=1}^{|\mathcal{G}|}$. Specifically, the decision-making for server-side topology evolution relies on Bayesian hypothesis testing. It treats the current global prototype topology $\mathcal{G}$ as the component parameters of a BGMM. Each prototype cluster $\mathbf{g}_j$ corresponds to a Gaussian distribution component with mean $\mu_j$ and covariance $\mathbf{\Sigma}_j$. Then, a NIW conjugate prior is introduced for each component, as it simultaneously models both the mean and covariance of the Gaussian distribution. The prior is defined as:

$$
\begin{aligned}
p(\boldsymbol{\mu}_j, \mathbf{\Sigma}_j) &= \mathcal{NIW}(\boldsymbol{\mu}_j, \mathbf{\Sigma}_j | \mathbf{m}_0, \kappa_0, \nu_0, \mathbf{S}_0) \\
&= \mathcal{N}(\boldsymbol{\mu}_j | \mathbf{m}_0, \frac{1}{\kappa_0} \mathbf{\Sigma}_j) \cdot \mathcal{W}^{-1}(\mathbf{\Sigma}_j | \nu_0, \mathbf{S}_0)
\end{aligned}
\tag{3}
$$

where $\mathcal{W}^{-1}$ is the Inverse-Wishart distribution, $\mathbf{m}_0$ is the mean prior typically initialized to the current prototype cluster $\mathbf{g}_j$, $\kappa_0$ controls the strength of the mean prior, $\nu_0$ is the degrees of freedom which must be greater than the feature dimension $d - 1$, and $\mathbf{S}_0$ is the scale matrix.

For each existing prototype cluster $\mathbf{g}_j$ in the global prototype topology $\mathcal{G}$, the corresponding data distribution may not be uni-modal but may contain undiscovered substructures. Therefore, we hypothesize that the feature corresponding to the current prototype cluster $\mathbf{g}_j$ can be better represented by two finer-grained components. Specifically, we assume all points in the set $\mathcal{S}_j$ are from a single Gaussian distribution with unknown mean $\boldsymbol{\mu}$ and covariance $\mathbf{\Sigma}$. Then, we partition them closest to $\mathbf{g}_j$ into two sub-clusters and obtain the means $\boldsymbol{\mu}_{j,1}, \boldsymbol{\mu}_{j,2}$, covariances $\mathbf{\Sigma}_{j,1}, \mathbf{\Sigma}_{j,2}$, and the number of prototypes in each sub-cluster $N_{j,1}, N_{j,2}$ ($N_{j,1} + N_{j,2} = N_j$).

We then compute the Bayes Factor between the split hypothesis ($H_1$: the feature is better represented by two components) and the non-split hypothesis ($H_0$: the feature is better represented by one component) by comparing the marginal likelihoods under the two hypotheses. For hypothesis $H_0$:

$$
p(\mathcal{S}_j | H_0) = \int \int \left[ \prod_{i=1}^{N_j} \mathcal{N}(\mathbf{s}_i | \boldsymbol{\mu}, \mathbf{\Sigma}) \right] \mathcal{NIW}(\boldsymbol{\mu}, \mathbf{\Sigma} | \mathbf{m}_0, \kappa_0, \nu_0, \mathbf{S}_0) \, d\boldsymbol{\mu} \, d\mathbf{\Sigma}
\tag{4}
$$

For hypothesis $H_1$:

$$
p(\mathcal{S}_j | H_1) = p(\mathcal{S}_{j,1} | H_0) \cdot p(\mathcal{S}_{j,2} | H_0)
\tag{5}
$$

Through the NIW conjugate prior, the above marginal likelihood has an analytical solution. The log marginal likelihood under a single component is:

$$
\begin{aligned}
\log p(\mathcal{S}_j | H_0) = &- \frac{N_j d}{2} \log(2\pi) + \frac{d}{2} \log(\frac{\kappa_0}{\kappa_{N_j}}) + \log \Gamma_d(\frac{\nu_{N_j}}{2}) - \log \Gamma_d(\frac{\nu_0}{2}) \\
&+ \frac{\nu_0}{2} \log |\mathbf{S}_0| - \frac{\nu_{N_j}}{2} \log |\mathbf{S}_{N_j}|
\end{aligned}
\tag{6}
$$

where $\Gamma_d$ is the multivariate Gamma function, $\kappa_{N_j} = \kappa_0 + N_j$, $\nu_{N_j} = \nu_0 + N_j$, and $\mathbf{S}_{N_j}$ is the posterior scale matrix calculated as:

$$
\mathbf{S}_{N_j} = \mathbf{S}_0 + \mathbf{S}_{\bar{\mathbf{s}}} + \frac{\kappa_0 N_j}{\kappa_0 + N_j} (\bar{\mathbf{s}} - \mathbf{m}_0)(\bar{\mathbf{s}} - \mathbf{m}_0)^T
\tag{7}
$$

Here, $\bar{\mathbf{s}}$ is the mean of the points $\mathcal{S}_j$, and $\mathbf{S}_{\bar{\mathbf{s}}}$ is the scatter matrix $\sum_{i=1}^{N_j}(\mathbf{s}_i - \bar{\mathbf{s}})(\mathbf{s}_i - \bar{\mathbf{s}})^T$.

We compute the statistical evidence for splitting, i.e., the marginal likelihood ratio:

$$R_{\text{split}} = \frac{p(\mathcal{S}_j|H_1)}{p(\mathcal{S}_j|H_0)} = \frac{p(\mathcal{S}_{j,1}|H_0) \cdot p(\mathcal{S}_{j,2}|H_0)}{p(\mathcal{S}_j|H_0)} \tag{8}$$

If $R_{\text{split}} > 1$, there is considered to be statistical evidence supporting the splitting of prototype cluster $\mathbf{g}_j$ into two new prototype clusters $\mathbf{g}_{j,1}$ and $\mathbf{g}_{j,2}$. The old prototype cluster $\mathbf{g}_j$ is removed from $\mathcal{G}$, the new prototype clusters are added, and the global prototype cluster count $|\mathcal{G}|$ increases by 1.

Similarly, we need to check whether two prototype clusters are too close and should be merged into one to eliminate redundancy and noise. For each pair of adjacent prototype clusters $(\mathbf{g}_j, \mathbf{g}_l)$ in the topology $\mathcal{G}$, we suspect they may represent the same semantic concept. Therefore, we compute the Bayes Factor between the merge hypothesis ($H_1$: the feature corresponding to the two prototype clusters comes from the same component) and the non-merge hypothesis ($H_0$: the feature comes from two different components). For hypothesis $H_1$:

$$p(\mathcal{S}_j \cup \mathcal{S}_l|H_1) = \int \int \left[ \prod_{i=1}^{N_j+N_l} \mathcal{N}(\mathbf{s}_i|\boldsymbol{\mu}, \boldsymbol{\Sigma}) \right] \mathcal{NIW}(\boldsymbol{\mu}, \boldsymbol{\Sigma}|\mathbf{m}_0, \kappa_0, \nu_0, \mathbf{S}_0) \, d\boldsymbol{\mu} \, d\boldsymbol{\Sigma} \tag{9}$$

For hypothesis $H_0$:

$$p(\mathcal{S}_j \cup \mathcal{S}_l|H_0) = p(\mathcal{S}_j|H_0) \cdot p(\mathcal{S}_l|H_0) \tag{10}$$

The statistical evidence for merging is the marginal likelihood ratio:

$$R_{\text{merge}} = \frac{p(\mathcal{S}_j \cup \mathcal{S}_l|H_1)}{p(\mathcal{S}_j \cup \mathcal{S}_l|H_0)} = \frac{p(\mathcal{S}_j \cup \mathcal{S}_l|H_1)}{p(\mathcal{S}_j|H_0) \cdot p(\mathcal{S}_l|H_0)} \tag{11}$$

If $R_{\text{merge}} > 1$, there is considered to be strong evidence supporting the merging of prototype clusters $\mathbf{g}_j$ and $\mathbf{g}_l$ into a new prototype cluster $\mathbf{g}_{\text{new}}$. The parameters of the new prototype cluster can be set as the weighted average of the two subsets: $\mathbf{g}_{\text{new}} = (N_j\mathbf{g}_j + N_l\mathbf{g}_l)/(N_j + N_l)$. The old prototype clusters are removed, the new prototype cluster is added, and the global prototype cluster count $|\mathcal{G}|$ decreases by 1.

## 3.2 STABILIZED EVOLUTION WITH GRADUAL CONSTRAINTS

During the evolution of the prototype topology, purely evidence-based split and merge operations may lead to unstable structures. For example, when a split operation produces two sub-clusters with significantly different sizes, or when two spatially close but semantically distinct prototype clusters are merged due to numerical coincidence, the stability of the global topology and semantic consistency can be negatively affected. To address this issue, we introduce a penalty mechanism to constrain the split decisions.

Specifically, for a split operation, we define the balance ratio $B$ as:

$$B = \frac{\min(N_{j,1}, N_{j,2})}{\max(N_{j,1}, N_{j,2})} \tag{12}$$

where $N_{j,1} + N_{j,2} = N_j$. A value of $B$ closer to 1 indicates a more balanced split, while one closer to 0 indicates a more imbalanced split. Based on this, we define the split imbalance penalty as:

$$P_{\text{split}} = \beta_{\text{split}} \cdot (1 - B) \tag{13}$$

where $\beta_{\text{split}}$ is a hyperparameter controlling the strength of the penalty. The penalized split decision criterion becomes:

$$\ln(R_{\text{split}}) - P_{\text{split}} > 0 \tag{14}$$

A split operation is executed only if the penalized evidence strength remains greater than 0, which effectively suppresses the generation of low-quality cluster structures caused by imbalanced splits.

For a merge operation, we focus on the semantic relevance between the two candidate prototype clusters $\mathbf{g}_j$ and $\mathbf{g}_l$. Even if they are spatially close, they should not be merged if their semantic

meanings differ significantly. To this end, we introduce a merge penalty based on feature distributions corresponding to the two prototype clusters:

$$D(\mathbf{p}_j||\mathbf{p}_l) = \frac{1}{2}D_{\text{KL}}(\mathbf{p}_j||\mathbf{m}) + \frac{1}{2}D_{\text{KL}}(\mathbf{p}_l||\mathbf{m}) \tag{15}$$

where $\mathbf{m} = \frac{1}{2}(\mathbf{p}_j + \mathbf{p}_l)$ and $D_{\text{KL}}$ is the Kullback-Leibler divergence. The merge penalty can then be defined as:

$$P_{\text{merge}} = \beta_{\text{merge}} \cdot D(\mathbf{p}_j||\mathbf{p}_l) \tag{16}$$

where $\beta_{\text{merge}}$ is a hyperparameter controlling the strength of the merge penalty. In the merge decision, we apply the penalty as:

$$\ln(R_{\text{merge}}) - P_{\text{merge}} > 0 \tag{17}$$

This criterion ensures that a merge operation is executed only when both statistical evidence supports the merge and the semantic meanings of the two prototype clusters are sufficiently similar.

By introducing the imbalance penalties $P_{\text{split}}$ and $P_{\text{merge}}$, we impose necessary constraints on the topology evolution process. This ensures that the decision-making process relies not only on statistical likelihood but also considers the quality of the cluster structures and semantic consistency, providing clients with higher-quality anchors in the feature space.

### 3.3 DUAL-OBJECTIVE LOSS WITH TOPOLOGY CONTRAST

After the server updates the global prototype topology $\mathcal{G}$, it distributes their cluster centers to clients, providing them with consensus-based anchors in the feature space. These anchors can be used to align and optimize the feature representations of clients' local models. Specifically, the local training objective for client $k$ including a cross-entropy loss and a prototype topology-aware contrastive loss.

The cross-entropy loss ensures the model's basic classification capability by measuring the discrepancy between the model's predictions and the true labels:

$$\mathcal{L}_{\text{CE}} = -\frac{1}{N_k}\sum_{i=1}^{N_k}\log\frac{\exp(o_i^{[y_i]})}{\sum_{j=1}^{M}\exp(o_i^{[j]})} \tag{18}$$

where $o_i$ is the output logits of the model for sample $\mathbf{x}_i$, and $[y_i]$ indexes the ground-truth class.

The prototype topology-aware contrastive loss drives the alignment of local feature representations toward the global consensus, enhancing their discriminability and cross-domain invariance. The core idea is that in the feature space, a sample's feature should be close to the prototype cluster centers of its own class and far from prototype cluster centers of all other classes (van den Oord et al., 2018). For a sample $(\mathbf{x}_i, y_i)$ on client $k$, the positive prototype cluster centers $\mathcal{G}^+$ for its feature $\mathbf{z}_i$ consists of all prototype cluster centers in $\mathcal{G}$ with label $y_i$. The negative prototype cluster centers $\mathcal{G}^-$ includes all prototype cluster centers in $\mathcal{G}$ whose labels are not $y_i$. The prototype topology-aware contrastive loss is defined as:

$$\mathcal{L}_{\text{contra}} = -\frac{1}{N_k}\sum_{i=1}^{N_k}\log\frac{\sum_{\mathbf{g}^+\in\mathcal{G}^+}\exp(\text{sim}(\mathbf{z}_i, \mathbf{g}^+)/\tau)}{\sum_{\mathbf{g}^+\in\mathcal{G}^+}\exp(\text{sim}(\mathbf{z}_i, \mathbf{g}^+)/\tau) + \sum_{\mathbf{g}^-\in\mathcal{G}^-}\exp(\text{sim}(\mathbf{z}_i, \mathbf{g}^-)/\tau)} \tag{19}$$

where $\text{sim}(\cdot, \cdot)$ is a similarity function, typically cosine similarity. The temperature hyperparameter $\tau > 0$ controls the sharpness of the distribution. This encourages the feature $\mathbf{z}_i$ to exhibit high similarity with all positive centers and low similarity with all negative centers. Since the alignment targets are the high-quality global semantic centers optimized by the server, and the numerator sums over all positive centers, a sample is allowed to be close to any subclass center within its own class.

The overall training objective for client $k$ is to minimize the following joint loss function:

$$\mathcal{L}_{\text{local}} = \mathcal{L}_{\text{CE}} + \lambda \cdot \mathcal{L}_{\text{contra}} \tag{20}$$

where $\lambda$ is a hyperparameter that balances the importance of the supervised loss and the contrastive loss. By jointly optimizing these two objectives, the local model not only learns discriminative classification boundaries but also aligns its feature representations with the globally consistent semantic topology, thereby significantly improving the model's generalization capability. After local training, the client computes feature cluster centers for each class based on its local data to form local prototypes, which are then uploaded to the server to facilitate the next round of topology evolution.

Table 1: Main results of performance comparison on Digit with 5 clients where "Avg." means average results among all clients. The best results are shown in bold.

| Methods | Models | MNIST | SVHN | USPS | Synth | MNIST-M | Avg. |
|---------|--------|-------|------|------|-------|---------|------|
| FedOPT | ResNet-10 | 88.75±0.07 | 26.00±0.89 | 82.58±1.28 | 43.50±1.68 | 56.42±0.69 | 59.45 |
| FedDyn | ResNet-10 | 97.62±0.05 | 71.78±0.10 | 96.51±0.12 | 87.73±0.03 | 85.16±0.34 | 87.76 |
| Moon | ResNet-10 | 96.45±0.02 | 62.10±0.40 | 94.03±0.18 | 81.70±0.21 | 81.11±0.36 | 83.08 |
| FedProto | ResNet-10 | 97.65±0.04 | 72.02±0.50 | 96.20±0.22 | 87.36±0.16 | 84.36±0.05 | 87.52 |
| FPL | ResNet-10 | 98.10±0.05 | 77.02±1.09 | 96.99±0.08 | 90.50±0.34 | 87.89±0.26 | 90.10 |
| FedPLVM | ResNet-10 | 97.88±0.10 | 81.15±0.30 | 96.49±0.08 | 92.08±0.09 | 90.17±0.20 | 91.55 |
| FedHEAL | ResNet-10 | 97.89±0.07 | 59.80±0.61 | 96.29±0.15 | 83.60±0.45 | 85.56±0.46 | 84.63 |
| FedPall | ResNet-50 | 97.99±0.38 | 73.70±1.21 | 91.88±0.70 | 96.37±0.19 | 77.48±0.05 | 87.48 |
| MPFT | CLIP | 91.66±0.08 | 41.92±0.43 | 84.00±0.13 | 75.48±0.12 | 68.31±0.14 | 72.27 |
| FedPTE | CLIP | **95.31±0.03** | **46.68±0.11** | **93.55±0.08** | **81.42±0.06** | **71.13±0.02** | **77.62** |
| FedPTE | ResNet-10 | **98.88±0.07** | **84.93±0.17** | **98.32±0.07** | **95.13±0.07** | **92.65±0.09** | **93.98** |

# 4 EXPERIMENTS

## 4.1 EXPERIMENT SETUP

We conduct experimental evaluations on Digit (Zhou et al., 2020) and Office (Gong et al., 2012), where Digit includes 5 domains: MNIST (M), SVHN (SV), USPS (U), Synth (SY), MNIST-M (M-M), and Office includes four domains: Amazon, Caltech, DSLR, Webcam. Following the settings of existing methods, we use ResNet-10 (He et al., 2016) as the local model for clients and compare with FedOPT (Reddi et al., 2021), FedDyn (Acar et al., 2021), Moon (Li et al., 2021), FedProto (Tan et al., 2022), FPL (Huang et al., 2023), FedPLVM (Wang et al., 2024), FedHEAL (Chen et al., 2024b), FedPall (Zhang et al., 2025c), which uses ResNet-50 and a three-layer multilayer perceptron. Meanwhile, we use CLIP-ViT-B-32 (Radford et al., 2021) as the local model for comparison with MPFT (Zhang et al., 2025a). Convergence analysis is in Appendices A, additional experiments on different domain distributions and medical datasets are presented in Appendices B. Supplementary experimental details can be found in Appendices C. Further analysis of the method and the disclosure of LLM use are provided in Appendices D and E, respectively.

Specifically, unless otherwise stated, each client possesses the entire dataset of one domain, with local training conducted for 2 epochs and a batch size of 512. For Digit, clients undergo 50 communication rounds, while for Office, they undergo 80 communication rounds. For methods using ResNet, we adopt a learning rate of 1e-2, whereas for methods using CLIP-ViT-B-32, we use a learning rate of 1e-3. In FedPTE, we set $\lambda = 100$, $\tau = 0.06$, $\beta_{\text{split}} = 1.0$, and $\beta_{\text{merge}} = 1.5$. All experiments are repeated three times with different random seeds on a single Tesla 40G A100 GPU.

## 4.2 MAIN RESULTS

**Domain Shift Results.** Tables 1 and 2 present the performance comparison between FedPTE and baselines on Digit and Office, respectively. Among them, FedOPT is prone to being dominated by gradients from a single domain under strong domain shift scenarios, leading to significant performance degradation on domains that exhibit substantial distributional discrepancies from the mainstream, such as SVHN, Synth, and Amazon. In contrast, methods incorporating dynamic regularization or prototype clustering, such as FedDyn, FedProto, and FedPLVM, demonstrate better stability in representation learning and overall outperform simple aggregation approaches.

Additionally, the alignment between the pre-trained model and the target domain significantly influences the results. CLIP-based methods achieve remarkable performance on Office, which closely resembles CLIP's pre-training distribution, but perform considerably worse on Digit, a dataset comprising low-resolution and stylistically diverse digit images. This indicates that when pre-trained representations mismatch the distribution of the downstream domain, fine-tuning is susceptible to performance degradation due to noise in the uploaded prototypes.

In comparison, FedPTE achieves consistent improvements, which lies in its evidence-driven maintenance of prototype topology and topology-aware client optimization, collectively enhancing in-class

Table 2: Main results of performance comparison on Office with 4 clients where "Avg." means average results among all clients. The best results are shown in bold.

| Methods | Models | Amazon | Caltech | DSLR | Webcam | Avg. |
|---------|--------|--------|---------|------|--------|------|
| FedOPT | ResNet-10 | 23.26±1.07 | 13.19±0.21 | 33.33±1.47 | 41.81±2.11 | 27.90 |
| FedDyn | ResNet-10 | 63.54±2.55 | 46.37±0.55 | 42.71±2.95 | 63.28±1.60 | 53.97 |
| Moon | ResNet-10 | 55.90±0.89 | 36.59±0.42 | 63.54±1.47 | 78.53±0.80 | 58.64 |
| FedProto | ResNet-10 | 50.17±3.54 | 38.37±0.21 | 61.46±1.47 | 75.14±0.80 | 56.29 |
| FPL | ResNet-10 | 60.59±5.35 | 45.93±1.79 | 40.62±4.42 | 61.02±0.03 | 52.04 |
| FedPLVM | ResNet-10 | 75.12±1.13 | 52.22±2.21 | 65.75±4.42 | 78.36±2.77 | 67.86 |
| FedHEAL | ResNet-10 | 68.85±1.88 | 46.97±4.03 | 43.33±4.78 | 76.33±2.85 | 58.87 |
| FedPall | ResNet-50 | 76.21±1.59 | 51.41±4.20 | 66.67±1.86 | 67.82±1.99 | 65.53 |
| MPFT | CLIP | 91.30±0.11 | 91.67±0.23 | 96.88±0.07 | 96.47±0.19 | 94.08 |
| FedPTE | CLIP | **97.92±0.05** | **96.44±0.12** | **100.00±0.02** | **100.00±0.04** | **98.59** |
| FedPTE | ResNet-10 | **80.21±0.43** | **57.38±0.91** | **71.79±3.90** | **82.66±1.38** | **73.01** |

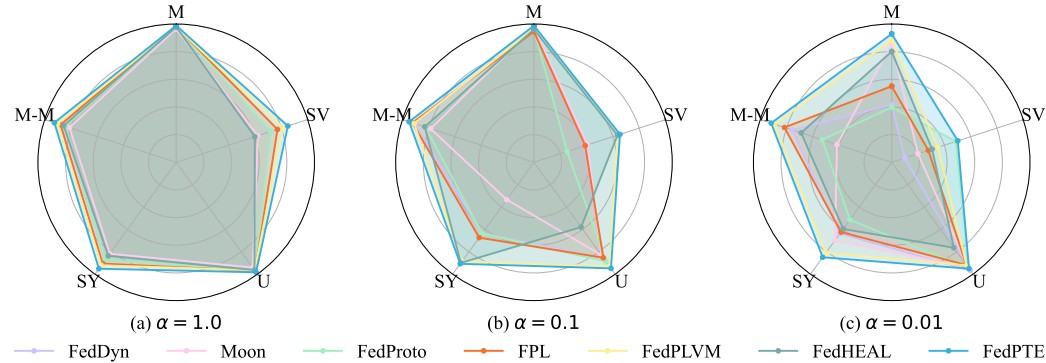

(a) $\alpha = 1.0$      (b) $\alpha = 0.1$      (c) $\alpha = 0.01$

FedDyn   Moon   FedProto   FPL   FedPLVM   FedHEAL   FedPTE

Figure 2: Main results on Digit with 5 clients under different Dirichlet partitions $\alpha$.

separability in the representation space and reducing cross-domain representation variance. As a result, FedPTE demonstrates robust performance advantages, proving the effectiveness of dynamic evidence-based prototype topology and topology-aware alignment in mitigating domain shift.

**Domain Shift Results with Non-IID Data.** We explore the impact of Non-IID data on model performance under domain shift, as shown in Figure 2. Here, a smaller $\alpha$ indicates higher data heterogeneity. As heterogeneity increases, baselines exhibit significant performance degradation on complex domains such as SVHN and USPS, indicating their difficulty in simultaneously addressing the dual challenges of domain shift and data heterogeneity. In contrast, FedPTE outperforms all comparative methods across all domains. Through its evidence-driven prototype topology maintenance mechanism, it maintains stable in-class separability even under highly heterogeneous data distributions, demonstrating its effectiveness and cross-domain generalization capability.

### 4.3 ABLATION RESULTS

**Component Ablation.** We explore the impact of different components on the performance of FedPTE, as shown in Table 3. When only splitting $R_{\text{split}}$ is used, the model's average accuracy increases by 2.05%, demonstrating the necessity of dynamically adjusting the number of prototype clusters. After introducing the penalty term for splitting $P_{\text{split}}$, the performance further improves to 86.31%. When merging $R_{\text{merge}}$ and its cor-

Table 3: Ablation results on Digit with 5 clients.

| $R_{\text{split}}$ | $P_{\text{split}}$ | $R_{\text{merge}}$ | $P_{\text{merge}}$ | $\mathcal{L}_{\text{contra}}$ | Avg. |
|--------|--------|--------|--------|--------|------|
| | | | | | 83.67 |
| ✓ | | | | | 85.72 |
| ✓ | ✓ | | | | 86.31 |
| ✓ | ✓ | ✓ | | | 88.14 |
| ✓ | ✓ | ✓ | ✓ | | 89.43 |
| ✓ | ✓ | ✓ | ✓ | ✓ | **93.98** |

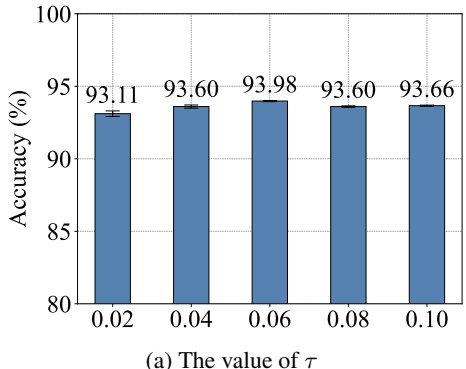 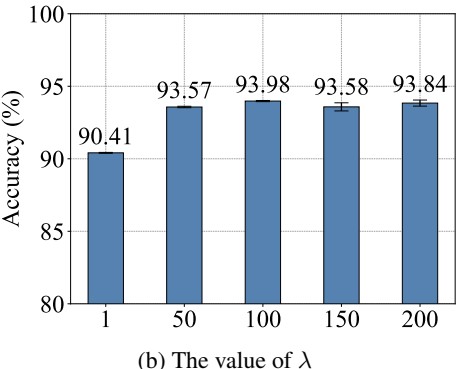

(a) The value of $\tau$          (b) The value of $\lambda$

Figure 3: Hyper-parameter ablation results on Digit with 5 clients.

responding penalty term $P_{\text{merge}}$ are incorporated, the accuracy reaches 88.14% and 89.43%, respectively, indicating that a reasonable merging mechanism effectively optimizes the prototype topology. Finally, with the introduction of prototype topology-aware contrastive loss $\mathcal{L}_{\text{contra}}$, the model achieves its best performance, highlighting the critical role of contrastive learning in enhancing feature separability and cross-domain generalization capability. These results validate the effectiveness and necessity of each component in FedPTE.

**Hyper-parameter Ablation.** Figure 3 shows the impact of temperature coefficient $\tau$ and loss weighting factor $\lambda$ on the performance of FedPTE. When $\tau$ increases from 0.02 to 0.06, the average accuracy improves steadily from 93.11% to 93.98%, with the best performance and the lowest standard deviation achieved at $\tau = 0.06$, indicating the most stable model performance under this configuration. As $\tau$ further increases to 0.10, the accuracy slightly decreases to 93.60%. Overall, FedPTE exhibits relatively low sensitivity to changes in the temperature coefficient.

Additionally, when $\lambda = 1$, the model performance is the lowest, suggesting insufficient strength of contrastive learning. As $\lambda$ increases to 100, the accuracy significantly improves to 93.98%, confirming the importance of contrastive learning in the training process. When $\lambda$ is further increased to 200, the accuracy slightly decreases to 93.84% with an increased standard deviation. These results demonstrate that FedPTE achieves its optimal configuration at $\tau = 0.06$ and $\lambda = 100$, where the model effectively leverages contrastive learning to enhance feature separability while maintaining training stability, thereby delivering the best performance.

Table 4: Multi-domain results of performance comparison on Office with various number of clients.

| Methods | Model | Mixed Ratio= 0.3 | | | Mixed Ratio= 0.5 | | | Mixed Ratio= 0.7 | | |
|---|---|---|---|---|---|---|---|---|---|---|
| | | 4 | 10 | 16 | 4 | 10 | 16 | 4 | 10 | 16 |
| FedOPT | ResNet-10 | 26.21 | 29.90 | 27.44 | 29.43 | 29.98 | 26.63 | 29.60 | 29.82 | 21.94 |
| FedDyn | ResNet-10 | 55.09 | 60.25 | 58.72 | 58.73 | 62.41 | 63.21 | 56.43 | 61.70 | 54.64 |
| Moon | ResNet-10 | 56.59 | 58.06 | 59.65 | 57.95 | 60.03 | 58.89 | 59.82 | 59.84 | 52.73 |
| FedProto | ResNet-10 | 51.22 | 55.80 | 55.46 | 52.48 | 56.61 | 57.82 | 55.01 | 51.10 | 50.87 |
| FPL | ResNet-10 | 57.67 | 63.63 | 61.58 | 58.84 | 64.33 | 63.34 | 58.86 | 66.66 | 54.76 |
| FedPLVM | ResNet-10 | 69.68 | 62.51 | 63.36 | 66.61 | 61.19 | 62.52 | 68.02 | 66.01 | 58.20 |
| FedHEAL | ResNet-10 | 71.82 | 68.85 | 60.89 | 66.96 | 66.94 | 62.97 | 67.82 | 60.55 | 59.73 |
| FedPTE | ResNet-10 | **72.64** | **75.62** | **74.72** | **73.28** | **72.48** | **70.23** | **73.20** | **73.15** | **72.48** |

## 4.4 MULTI-DOMAIN RESULTS

Inspired by more complex FL scenarios, we explore the case where a client's data is mixed with data from other domains, as well as the case where multiple clients share the same domain, as shown in Table 4. Here, the mixed ratio controls the proportion of data from other domains that a client possesses, and each domain is assigned to at least one client randomly. As the mixed ratio increases from 0.3 to 0.7, the performance of all baselines exhibits varying degrees of fluctuation

or degradation. This indicates that a high degree of inter-domain data mixing intensifies the challenges of local model training and global aggregation. In contrast, FedPTE demonstrates minimal performance fluctuation, highlighting the ability of its prototype topology evolution mechanism to effectively adapt to the complex multi-modal data distributions within clients.

Furthermore, as the number of clients increases from 4 to 16, the heterogeneity is further amplified. Most baselines show a declining performance trend with the growth in client number, particularly under a mixed ratio of 0.7 with 16 clients, where FedOPT's performance drops to 21.94%. In comparison, FedPTE maintains stability as the client scale expands, validating the strong scalability of its dynamic topology maintenance mechanism across different system sizes. It effectively captures local sub-class structures under multi-domain mixing while leveraging topology-aware contrastive learning to enhance in-class consistency and inter-class discriminability of features, thereby achieving robust feature alignment and model generalization.

Table 5: Unseen domain results of performance comparison on Digit with 5 clients.

| Methods | Self-training | | | | | None | | | | |
|---|---|---|---|---|---|---|---|---|---|---|
| | M | SV | U | SY | M-M | M | SV | U | SY | M-M |
| FedOPT | 80.61 | 29.03 | 70.38 | 31.99 | 45.46 | 80.05 | 22.32 | 53.01 | 33.89 | 48.07 |
| FedDyn | 96.26 | 58.09 | 89.95 | 75.02 | 62.79 | 95.53 | 50.91 | 84.73 | 72.43 | 59.11 |
| Moon | 94.74 | 51.53 | 86.61 | 70.03 | 62.21 | 93.63 | 43.17 | 81.51 | 66.48 | 57.76 |
| FedProto | 96.69 | 57.76 | 90.48 | 76.11 | 63.82 | 95.69 | 51.99 | 86.67 | 73.34 | 59.38 |
| FPL | 97.43 | 61.33 | 89.77 | 79.47 | 64.84 | 96.63 | 58.50 | 87.10 | 77.31 | 60.31 |
| FedPLVM | 96.11 | 65.03 | 90.06 | 80.40 | 68.74 | 96.09 | 60.27 | 88.11 | 80.20 | 65.64 |
| FedHEAL | 96.44 | 54.14 | 90.11 | 66.85 | 62.64 | 96.30 | 45.95 | 85.32 | 62.26 | 58.49 |
| FedPTE | **98.32** | **68.69** | **90.54** | **83.21** | **70.98** | **98.30** | **64.37** | **90.22** | **84.98** | **69.94** |

## 4.5 Unseen Domain Results

Table 5 presents the adaptation evaluation results of FedPTE and comparative methods on unseen domains. Following the setting in FDSE (Wang et al., 2025), we adopt a leave-one-domain-out evaluation protocol where each time one domain is held out as the target test domain while the remaining four domains participate in training. The model is evaluated on the target domain under two settings, including direct evaluation (None) and evaluation after self-training. For the self-training setup, we first use the model trained on the source clients to predict all data in the target domain and generate pseudo-labels. We then select the top 20% of samples with the highest pseudo-label confidence to form a high-confidence subset and perform 2 rounds of adaptive training on the model using this subset.

The experimental results demonstrate that FedPTE significantly outperforms all baselines under both evaluation settings, exhibiting superior domain generalization capability. In the direct evaluation, FedPTE achieves leading average performance across all five unseen domains, indicating that the learned feature representations possess stronger cross-domain invariance. After self-training, the performance of all methods improves, but FedPTE shows the most stable and robust enhancement. Notably, the performance fluctuation of FedPTE before and after self-training is minimal, suggesting that the initial model already provides a stable feature foundation, requiring only a small number of target domain samples for effective adaptation.

## 5 Conclusion

This paper addresses the issue of domain shift in FL by proposing a prototype topology evolution-based framework named FedPTE. It adaptively adjusts the global prototype topology through Bayesian inference and marginal likelihood ratios, effectively capturing multi-modal distribution characteristics within classes while reducing redundancy and noise. The stability constraint mechanism on the server effectively prevents training oscillations caused by abrupt topological changes, while the contrastive learning loss on clients further promotes the alignment of features with global semantic centers, enhancing the model's discriminability and generalization capability. Experiments validate the superior performance of FedPTE across multiple cross-domain tasks, demonstrating its ability to better model feature distributions in heterogeneous domains.

ACKNOWLEDGMENTS

This work was supported by the National Key R&D Program of China (2022YFB3103202).

ETHICS STATEMENT

This work adheres to the ICLR Code of Ethics. All authors have read and committed to upholding the code throughout the submission and review process. Our research presents a FL framework designed to improve model performance under domain shift without centralizing private client data. All experiments are conducted on publicly available benchmark datasets and curated medical imaging datasets that are de-identified and intended for research use. No new data involving human subjects is collected for this study.

Like all machine learning models, the performance of FedPTE is dependent on the training data. While we demonstrate its effectiveness across diverse domains, the model could potentially inherit or amplify biases present in the source datasets. We have evaluated FedPTE on datasets with inherent domain shifts to mitigate biases related to specific data sources and promote fairness. We believe the benefits of FedPTE, which aims to make FL more robust and applicable to real-world heterogeneous environments, outweigh the potential risks, which we consider to be minimal and manageable.

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

# A    CONVERGENCE ANALYSIS

Let $F(\mathbf{w}) = \frac{1}{K}\sum_{k=1}^{K}\mathcal{L}_k(\mathbf{w})$ be the global objective function, where $\mathcal{L}_k(\mathbf{w})$ is the local objective for client $k$, defined as:

$$\mathcal{L}_k(\mathbf{w}) = \mathcal{L}_{\text{CE}}(\mathbf{w}) + \lambda \cdot \mathcal{L}_{\text{contra}}(\mathbf{w}, \mathcal{G}) \tag{21}$$

We make the following assumptions:

**Assumption 1.** *Lipschitz Smoothness: The local objective $\mathcal{L}_k(\mathbf{w}, \mathcal{G})$ is $L$-smooth:*

$$\|\nabla\mathcal{L}_k(\mathbf{w}, \mathcal{G}) - \nabla\mathcal{L}_k(\mathbf{v}, \mathcal{G})\| \leq L\|\mathbf{w} - \mathbf{v}\|, \quad \forall\mathbf{w}, \mathbf{v}, \forall k \tag{22}$$

**Assumption 2.** *Bounded Variance: The stochastic gradient on each client has bounded variance:*

$$\mathbb{E}\left[\|\nabla\mathcal{L}_k(\mathbf{w}, \mathcal{G}; \xi) - \nabla\mathcal{L}_k(\mathbf{w}, \mathcal{G})\|^2\right] \leq \sigma^2 \tag{23}$$

**Assumption 3.** *Bounded Gradient: The expected squared norm of stochastic gradients is bounded:*

$$\mathbb{E}\left[\|\nabla\mathcal{L}_k(\mathbf{w}, \mathcal{G}; \xi)\|^2\right] \leq G^2 \tag{24}$$

**Assumption 4.** *Prototype Stability: The global prototype topology $\mathcal{G}$ evolves gradually such that:*

$$\|\mathcal{G}^{(t+1)} - \mathcal{G}^{(t)}\| \leq \delta \tag{25}$$

*where $\delta$ is a small constant, ensured by the penalty constraints in FedPTE.*

**Assumption 5.** *Lipschitz Contrastive Loss: The contrastive loss is $L_c$-Lipschitz with respect to prototype positions:*

$$|\mathcal{L}_{contra}(\mathbf{w}, \mathcal{G}) - \mathcal{L}_{contra}(\mathbf{w}, \mathcal{G}')| \leq L_c\|\mathcal{G} - \mathcal{G}'\| \tag{26}$$

*Proof.* Let $\mathbf{w}^{(t)}$ denote the global model at communication round $t$, and $\mathbf{w}_k^{(t+1)}$ is the local model of client $k$ after local updates.

For each client $k$, after $\tau$ steps of local SGD with learning rate $\eta$, by the $L$-smoothness of $\mathcal{L}_k$ and the bounded variance assumption, we have:

$$\mathbb{E}[\mathcal{L}_k(\mathbf{w}_k^{(t+1)}, \mathcal{G}^{(t)})] \leq \mathcal{L}_k(\mathbf{w}^{(t)}, \mathcal{G}^{(t)}) - \eta\tau\left(1 - \frac{L\eta\tau}{2}\right)\|\nabla\mathcal{L}_k(\mathbf{w}^{(t)}, \mathcal{G}^{(t)})\|^2 + \frac{L\eta^2\tau}{2}\sigma^2 \tag{27}$$

The contrastive loss $\mathcal{L}_{\text{contra}}$ depends on the global prototype set $\mathcal{G}$. Under Assumptions 4 and 5, the change in the contrastive loss component between rounds is bounded:

$$|\mathcal{L}_{\text{contra}}(\mathbf{w}, \mathcal{G}^{(t+1)}) - \mathcal{L}_{\text{contra}}(\mathbf{w}, \mathcal{G}^{(t)})| \leq \lambda L_c\delta \tag{28}$$

After local updates, the global model is aggregated as:

$$\mathbf{w}^{(t+1)} = \frac{1}{K}\sum_{k=1}^{K}\mathbf{w}_k^{(t+1)} \tag{29}$$

Due to the $L$-smoothness of the global objective $F$ and the bounded prototype evolution, we can derive:

$$\mathbb{E}[F(\mathbf{w}^{(t+1)}, \mathcal{G}^{(t+1)})] \leq F(\mathbf{w}^{(t)}, \mathcal{G}^{(t)}) - \eta\tau\left(1 - \frac{L\eta\tau}{2}\right)\|\nabla F(\mathbf{w}^{(t)}, \mathcal{G}^{(t)})\|^2 + \frac{L\eta^2\tau}{2}\sigma^2 + \lambda L_c\delta \tag{30}$$

Summing over $T$ rounds and rearranging, we obtain:

$$\frac{1}{T}\sum_{t=1}^{T}\|\nabla F(\mathbf{w}^{(t)}, \mathcal{G}^{(t)})\|^2 \leq \frac{2(F(\mathbf{w}^{(0)}, \mathcal{G}^{(0)}) - F^*)}{\eta\tau T(2 - L\eta\tau)} + \frac{L\eta\sigma^2}{2 - L\eta\tau} + \frac{2\lambda L_c\delta}{\eta\tau(2 - L\eta\tau)} \tag{31}$$

where $F^*$ is the minimum value of $F$.

By choosing $\eta = \mathcal{O}(1/\sqrt{T})$ and assuming $\delta = \mathcal{O}(1/\sqrt{T})$ (ensured by FedPTE's penalty mechanisms), we have:

$$\frac{1}{T}\sum_{t=1}^{T}\|\nabla F(\mathbf{w}^{(t)})\|^2 \leq \mathcal{O}\left(\frac{1}{\sqrt{T}}\right) \tag{32}$$

This shows that FedPTE converges to a stationary point at an average rate of $\mathcal{O}(1/\sqrt{T})$. The penalty mechanisms in FedPTE ensure $\delta$ is sufficiently small, thereby guaranteeing stable convergence. $\quad\square$

# B ADDITIONAL EXPERIMENTS

## B.1 MULTI-CLIENT RESULTS

**Domain-balanced.** Figure 4a shows the performance of FedPTE and baselines under domain-balanced settings with varying numbers of clients. As the number of clients increases, the accuracy of FedPTE slightly decreases from 93.98% to 91.92%, yet it consistently and significantly outperforms all baselines. This trend indicates that in domain-balanced environments with relatively uniform data distribution, FedPTE can effectively capture the feature diversity introduced by multiple client participations through its Bayesian evidence-driven prototype topology evolution mechanism, thereby maintaining high performance. Although an increased number of clients may introduce greater data heterogeneity, FedPTE's dynamic prototype adjustment and stability constraints ensure the robustness of the training process, preventing severe performance fluctuations. This trend demonstrates that FedPTE can effectively leverage the data diversity from multiple client participations, thereby achieving robust generalization in heterogeneous data environments.

**Domain-imbalanced.** We also explore the performance of FedPTE and baselines under domain-imbalanced settings, as shown in Figure 4b. Specifically, we configure at least half of the clients to be in the same domain, with the remaining clients proportionally distributed across the other domains. When at least half of the clients reside in the same domain, the data distribution exhibits significant skewness. FedPTE's accuracy decreases from 88.66% to 84.37%, but the decline is relatively small, and its performance remains significantly superior to other methods. For instance, with 25 clients, FedPTE achieves an accuracy of 84.37%, while FedPLVM, FPL, and FedProto only achieve 79.37%, 64.84%, and 52.70%, respectively. This highlights FedPTE's strong adaptability under severe distribution shift. Its prototype topology-aware contrastive learning mechanism effectively aligns local features with global semantic centers. Concurrently, Bayesian inference automatically identifies redundant prototypes for merging and refines semantic substructures for splitting, thereby preserving semantic discriminability amidst inter-domain conflicts.

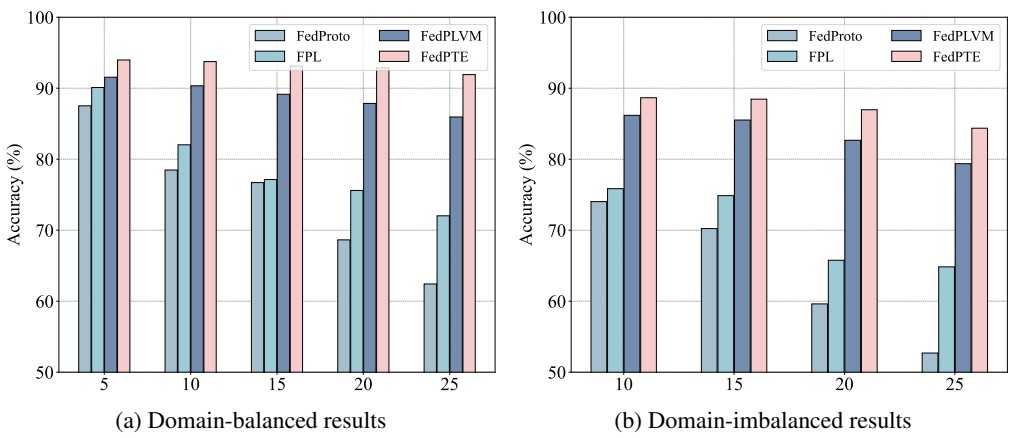

(a) Domain-balanced results      (b) Domain-imbalanced results

Figure 4: Multi-client results on Digit with various numbers of clients.

## B.2 REAL-WORLD DOMAIN RESULTS

Following the setup of FEAL (Chen et al., 2024a), we evaluate the performance of FedPTE on four real-world medical imaging datasets, including two classification datasets Fed-Camelyon, Fed-ISIC and two segmentation datasets Fed-Polyp, Fed-Prostate. Specifically, Fed-Camelyon is a breast cancer histology dataset comprising 5 different domains, Fed-ISIC is a skin lesion dataset with 4 domains, Fed-Polyp is an endoscopic polyp dataset with 4 domains, and Fed-Prostate is a prostate MRI dataset with 6 domains. Further details are provided in Table 6 and Figure 5. For the image classification task, we use ResNet-10 with 50 communication rounds. For the image segmentation task, we employ U-Net (Ronneberger et al., 2015) and change the cross-entropy loss into the Dice loss (Milletari et al., 2016) with 60 communication rounds.

Table 6: Details of real-world medical imaging datasets.

| Datasets | Train Size | Test Size | Source |
|---|---|---|---|
| Fed-Camelyon | 47548 | 11888 | Camelyon17 (Bándi et al., 2019) |
| | 27923 | 6981 | Camelyon17 (Bándi et al., 2019) |
| | 68043 | 17011 | Camelyon17 (Bándi et al., 2019) |
| | 103870 | 25968 | Camelyon17 (Bándi et al., 2019) |
| | 117377 | 29345 | Camelyon17 (Bándi et al., 2019) |
| Fed-ISIC | 9930 | 2483 | BCN (du Terrail et al., 2022) |
| | 3163 | 791 | HAM_vidir_molemax (du Terrail et al., 2022) |
| | 2691 | 672 | HAM_vidir_modern (du Terrail et al., 2022) |
| | 1807 | 452 | HAM_rosendahl (du Terrail et al., 2022) |
| Fed-Polyp | 800 | 200 | Kvasir (Jha et al., 2020) |
| | 157 | 39 | ETIS (Silva et al., 2014) |
| | 304 | 75 | ColonDB (Tajbakhsh et al., 2016) |
| | 490 | 122 | ClinicDB (Bernal et al., 2015) |
| Fed-Prostate | 225 | 36 | BIDMC (Litjens et al., 2014) |
| | 306 | 78 | BMC (Bloch et al., 2015) |
| | 134 | 24 | HK (Litjens et al., 2014) |
| | 387 | 81 | I2CVB (Lemaitre et al., 2015) |
| | 337 | 84 | RUNMC (Bloch et al., 2015) |
| | 152 | 23 | UCL (Litjens et al., 2014) |

Table 7: Real-world domain results of performance comparison on Fed-Polyp and Fed-Prostate. The best results are shown in bold.

| Methods | Fed-Polyp | | | | Fed-Prostate | | | | | |
|---|---|---|---|---|---|---|---|---|---|---|
| | Kvasir | ETIS | ColonDB | ClinicDB | BIDMC | BMC | HK | I2CVB | RUNMC | UCL |
| FedOPT | 51.62 | 58.21 | 55.81 | 59.25 | 50.04 | 50.31 | 50.90 | 50.86 | 50.53 | 50.43 |
| FedDyn | 77.54 | 62.46 | 73.11 | 63.30 | 83.01 | 83.86 | 86.90 | 80.24 | 80.49 | 79.72 |
| Moon | 49.87 | 56.35 | 58.94 | 54.63 | 85.82 | 78.51 | 86.16 | 81.44 | 78.34 | 77.31 |
| FedProto | 78.25 | 63.40 | 71.93 | 66.63 | 85.18 | 88.48 | 87.40 | 81.65 | 86.60 | 85.31 |
| FPL | 80.88 | 65.63 | 73.14 | 68.77 | 88.99 | 88.50 | 88.82 | 83.50 | 87.64 | 86.09 |
| FedPLVM | 76.05 | 64.08 | 66.44 | 68.77 | 88.24 | 89.28 | 87.99 | 87.49 | 86.75 | 83.50 |
| FedHEAL | 71.33 | 64.14 | 63.87 | 66.21 | 88.62 | 86.30 | 88.85 | 87.17 | 82.48 | 83.07 |
| FedPTE | **84.17** | **68.08** | **76.05** | **69.80** | **89.61** | **90.20** | **89.60** | **90.98** | **89.51** | **89.40** |

Specifically, Figure 6 presents the performance of FedPTE and baselines on medical image classification. The experimental results demonstrate that FedPTE achieves the best performance across all domains, showcasing its strong generalization capability in heterogeneous medical data scenarios. Its evidence-driven prototype topology evolution mechanism adaptively adjusts the global prototype structure to align with the data distribution characteristics of different medical centers, thereby enabling more effective cross-domain knowledge fusion.

In addition, Table 7 shows the performance of FedPTE and baselines on medical image segmentation, evaluated using the Dice coefficient (%), where higher values indicate better segmentation quality. The results indicate that FedPTE achieves the best performance across all ten test domains, demonstrating excellent domain generalization capability in pixel-level tasks. Notably, traditional FL methods perform significantly worse across all domains, further highlighting the necessity of explicitly handling domain shift in medical image segmentation.

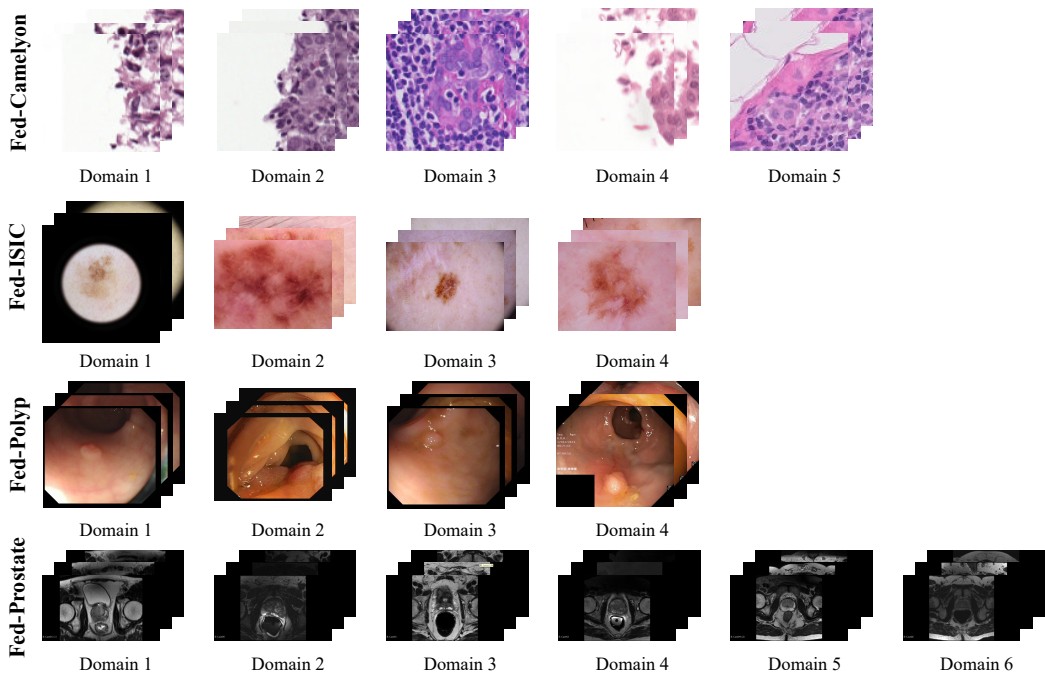

Figure 5: Examples from real-world medical imaging datasets.

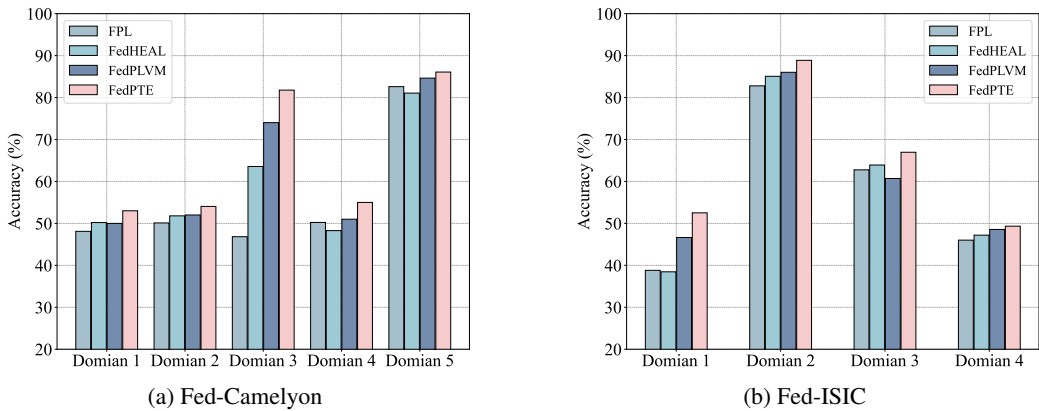

Figure 6: Real-world domain results of performance comparison on Fed-Camelyon and Fed-ISIC.

### B.3 NIW PRIOR STUDY

**Prior Hyper-parameter Ablation.** FedPTE employs fixed priors to make computations independent of each other and avoid maintaining complex states across communication rounds. Specifically, we set the strength of the mean prior $\kappa_0$ to 0.1, indicating weak confidence in the initial prior mean and allowing the posterior distribution to be more readily dominated by the client prototype data uploaded in the current round. For the degrees of freedom $\nu_0$, adhering to the theoretical requirements of the NIW distribution, we set it to $\nu_0 = d + 5$, where $d$ is the feature dimension. This is one of the minimum values that satisfies the distribution's definition while providing weak informativeness, again serving to avoid overly strong prior assumptions. For the scale matrix $\mathbf{S}_0$, we initialize it as $\mathbf{S}_0 = (\nu_0 - d - 1) \cdot I$, where $I$ is the identity matrix. This represents a unit-scaled covariance prior, which is a common and robust default choice.

Table 8: NIW Prior hyper-parameter ablation results on Digit with 5 clients.

| | $\kappa_0$ | | | | | $\nu_0$ | | | |
|---|---|---|---|---|---|---|---|---|---|
| | 0.01 | 0.05 | 0.1 | 0.5 | 1.0 | $d+2$ | $d+5$ | $d+10$ | $d+20$ |
| MNIST | 98.94 | 98.94 | 98.88 | 98.96 | 98.96 | 98.93 | 98.88 | 99.04 | 98.91 |
| SVHN | 84.41 | 84.46 | 84.93 | 84.52 | 84.65 | 83.93 | 84.93 | 84.66 | 84.20 |
| USPS | 98.28 | 98.49 | 98.32 | 98.44 | 98.28 | 98.33 | 98.32 | 98.55 | 98.55 |
| Synth | 94.97 | 95.11 | 95.13 | 95.22 | 95.05 | 95.04 | 95.13 | 95.12 | 95.12 |
| MNIST-M | 91.92 | 91.43 | 92.65 | 91.86 | 92.25 | 91.94 | 92.65 | 92.10 | 92.16 |
| Avg. | 93.71 | 93.69 | 93.98 | 93.80 | 93.84 | 93.63 | 93.98 | 93.89 | 93.79 |

Table 8 shows the impact of different prior hyperparameter values on model performance. The results indicate that as long as these parameters are set within reasonable ranges, which maintaining the nature of weakly informative priors, their variation has minimal impact on the final topology evolution decisions and model performance.

**Prior Component Ablation.** To explore the effectiveness of the NIW prior, we compared FedPTE's performance against variants using random evolution and a simple heuristic, as shown in Table 9. Specifically, the random evolution variant simulates undirected topological changes by performing splitting and merging operations at fixed probabilities.

Table 9: Prior component ablation results on Office with 4 clients. The best results are in bold.

| | Amazon | Caltech | DSLR | Webcam | Avg. |
|---|---|---|---|---|---|
| Random | 67.19 | 53.78 | 46.88 | 54.24 | 55.52 |
| Simple | 75.00 | 54.22 | 56.25 | 66.10 | 62.89 |
| NIW | **80.21** | **57.38** | **71.79** | **82.66** | **73.01** |

The simple heuristic variant removes the Bayesian estimation process for each cluster's covariance and discards the Bayesian posterior mean update mechanism. It replaces the log marginal likelihood used for split-merge decisions with the negative value of the cluster's sum of squared errors (SSE):

$$\log P(\mathcal{D}_k) \approx -\text{SSE}(\mathcal{D}_k) = -\sum_{\mathbf{x} \in \mathcal{D}_k} \|\mathbf{x} - \mu_k\|^2 \tag{33}$$

Consequently, the decision criterion shifts from selecting the operation that maximizes Bayesian evidence to the one that minimizes the total within-cluster SSE.

The experimental results demonstrate that the NIW prior is crucial for FedPTE's robust performance. Specifically, the random evolution strategy performs the worst since random operations frequently introduce semantically inconsistent or noisy topological changes. These arbitrary modifications disrupt the semantic structure of the prototype topology, failing to provide clients with stable and meaningful anchors in the feature space, ultimately leading to severe performance degradation. The simple heuristic variant lacks the comprehensive probabilistic perspective of Bayesian model comparison, which inherently balances goodness-of-fit with model complexity. Consequently, the SSE-based approach is more prone to getting stuck in local optima, which limits its ability to discover the globally optimal prototype topology.

In contrast, using the full NIW prior mechanism yields the best results, achieving an average accuracy of 73.01% and outperforming the other variants across all domains. This significant performance gap confirms that the Bayesian evidence-driven approach provides a more reliable and principled foundation for prototype topology evolution compared to alternatives lacking probabilistic grounding.

### B.4 PENALTY STUDY

**Penalty Ablation.** Figure 7 illustrates the impact of different penalty coefficient combinations on the performance of FedPTE. If the split penalty is too small, the model tends to over-split prototype clusters, generating a large number of redundant subclasses, which leads to fragmented in-class

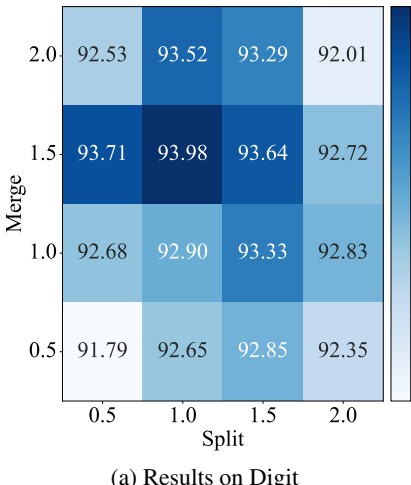
(a) Results on Digit

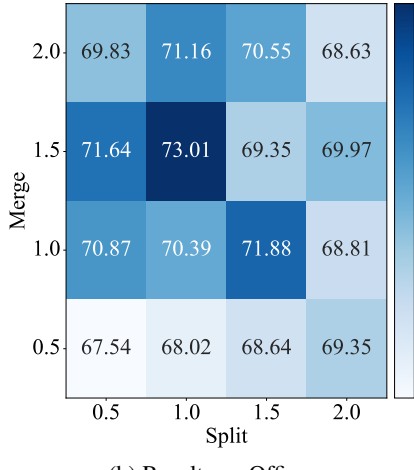
(b) Results on Office

Figure 7: Penalty ablation results on Digit with 5 clients and on Office with 4 clients.

structures and introduces noise. Conversely, the model fails to capture fine-grained semantic structures. Similarly, if the merge penalty is too small, the model excessively merges prototype clusters that are semantically similar but slightly different in distribution, while an excessively large merge penalty prevents necessary prototype aggregation, causing the topological structure to retain numerous similar yet unstable prototypes, thereby increasing the risk of training oscillations. Despite this, FedPTE exhibits a certain degree of robustness to the selection of penalty coefficients, maintaining relatively stable performance within the ranges of $\beta_{\text{split}} \in [0.5, 1.5]$ and $\beta_{\text{merge}} \in [1.0, 2.0]$. This enables it to effectively balance structural refinement and noise suppression in most scenarios, thereby enhancing the model's generalization capability and convergence stability.

Table 10: Adaptive weight results on Digit with 5 clients and on Office with 4 clients.

|  | Digit | | | | | Office | | | |
|---|---|---|---|---|---|---|---|---|---|
|  | MNIST | SVHN | USPS | Synth | MNIST-M | Amazon | Caltech | DSLR | Webcam |
| Adaptive | 98.89 | 85.25 | 98.55 | 95.57 | 93.47 | 81.77 | 60.89 | 65.62 | 77.97 |
| Fixed | 98.88 | 84.93 | 98.32 | 95.13 | 92.65 | 80.21 | 57.38 | 71.79 | 82.66 |

**Adaptive Weight Strategy.** We explore a strategy for adaptively adjusting the penalty coefficients $\beta_{\text{split}}$ and $\beta_{\text{merge}}$, as shown in Table 10. Specifically, we use the $S_{score}$ to determine whether to adaptively increase or decrease the coefficients:

$$S_{score} = 1 - \tanh\left(\frac{S_{\text{intra}}}{S_{\text{inter}}}\right) \tag{34}$$

where $S_{\text{intra}}$ represents the intra-cluster distance, i.e., the average distance from all points within a cluster to their center, and $S_{\text{inter}}$ denotes the inter-cluster distance, i.e., the average distance between all different cluster centers.

It can be observed that the adaptive penalty mechanism demonstrates more adaptable performance in most cases. When the intra-cluster distance is relatively small, $S_{score}$ approaches 1, and FedPTE appropriately relaxes the penalty constraints, allowing for more structural optimization. Conversely, when the intra-cluster structure is loose, it strengthens the penalty to avoid unstable topological changes. It is worth noting that on Office, the adaptive strategy performs better on the Amazon and Caltech domains, while the fixed parameter method still maintains an advantage on the DSLR and Webcam domains. This discrepancy may stem from the data distribution characteristics of different domains. When the data distribution is relatively compact, fixed penalty parameters can provide more stable constraints, whereas when the data distribution is more complex and variable, the adaptive penalty strategy can better handle such heterogeneity.

Overall, the adaptive penalty strategy shows promising potential. By dynamically balancing the flexibility of topological evolution and training stability, this mechanism can automatically adjust decision thresholds based on actual cluster structure characteristics, reducing the reliance on empirical hyperparameter tuning.

Table 11: Privacy protection results of various values of perturbation on Digit with 5 clients.

| Perturbation | MNIST | SVHN | USPS | Synth | MNIST-M | Avg. |
|---|---|---|---|---|---|---|
| 0.1 | 98.91 | 84.63 | 98.23 | 95.05 | 92.17 | 93.80 |
| 1.0 | 98.60 | 81.55 | 98.06 | 94.42 | 90.29 | 92.59 |
| 5.0 | 95.46 | 73.18 | 95.59 | 87.93 | 78.55 | 86.14 |
| 50.0 | 88.40 | 61.29 | 92.80 | 81.47 | 68.25 | 78.44 |

Table 12: Privacy protection results of various values of scale on Digit with 5 clients.

| Scale | MNIST | SVHN | USPS | Synth | MNIST-M | Avg. |
|---|---|---|---|---|---|---|
| 0.1 | 98.96 | 84.49 | 98.39 | 95.00 | 92.19 | 93.80 |
| 1.0 | 98.60 | 81.55 | 98.06 | 94.42 | 90.29 | 92.59 |
| 5.0 | 96.11 | 74.56 | 96.13 | 90.90 | 84.60 | 88.46 |
| 10.0 | 97.29 | 81.07 | 94.89 | 90.15 | 80.19 | 88.72 |

### B.5 PRIVACY PROTECTION

We explore the impact of incorporating differential privacy into prototypes on FedPTE, specifically examining the effects of different perturbation values and scale parameters, as shown in Tables 11 and 12. Following existing methods (Canonne et al., 2020; Wang et al., 2024), we set the privacy budget and the delta parameter to 4.0 and 1e-5, respectively. When the perturbation value is 0.1, the model maintains high performance, indicating that FedPTE can effectively preserve its feature representation capability under mild privacy protection. As the perturbation value increases to 5.0 and 50.0, the performance degrades significantly, with average accuracy dropping to 86.14% and 78.44%, respectively. This demonstrates that strong noise substantially interferes with the model's ability to learn features from complex domains.

Additionally, when the scale parameter is 0.1, the performance is similar to that with a perturbation value of 0.1. When the scale parameter increases to 5.0, the performance degradation is smaller compared to the case with a perturbation value of 5.0, suggesting that the scale parameter may provide a gentler noise control mechanism. The performance with a scale parameter of 10.0 is similar to that with 5.0, indicating that within this range, the model is relatively insensitive to changes in the scale parameter.

It is noteworthy that even when using privacy protection techniques, FedPTE achieves performance comparable to advanced baselines. As a result, FedPTE can maintain over 90% of its original performance under moderate privacy protection, effectively balancing the requirements of privacy preservation and model utility.

## C EXPERIMENTAL DETAILS

### C.1 PROTOTYPE TREND

Figure 8 illustrates the dynamic evolution trend of the prototype topology on the server across communication rounds. The results reveal that the number of prototypes per class fluctuates continuously between rounds, reflecting how the evidence-driven prototype evolution mechanism in FedPTE continuously optimizes the global feature representation. Further analysis shows a strong correlation between the variation in prototype count and the model's performance across different domains. By

comparing with the main experimental results in Figure 2, it can be observed that domains with a larger number of prototypes (such as MNIST and MNIST-M) generally correspond to higher classification accuracy. Moreover, the dynamic fluctuation in prototype count remains relatively stable, which is attributed to FedPTE's penalty mechanism that prevents excessive splitting or merging, thereby ensuring a balance between topological changes and semantic consistency.

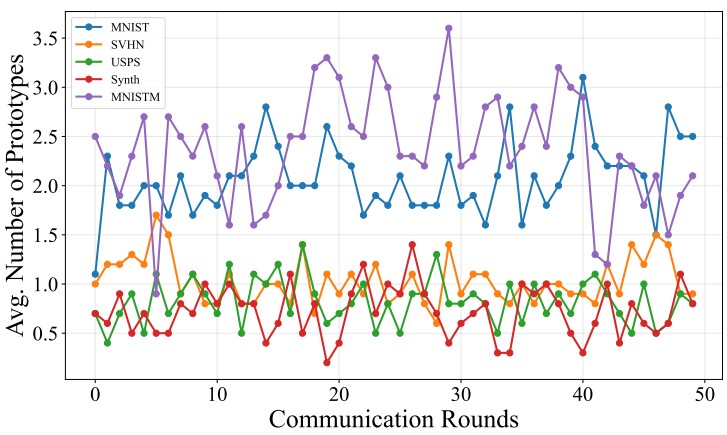

Figure 8: Prototype trend of FedPTE on Digit with 5 clients.

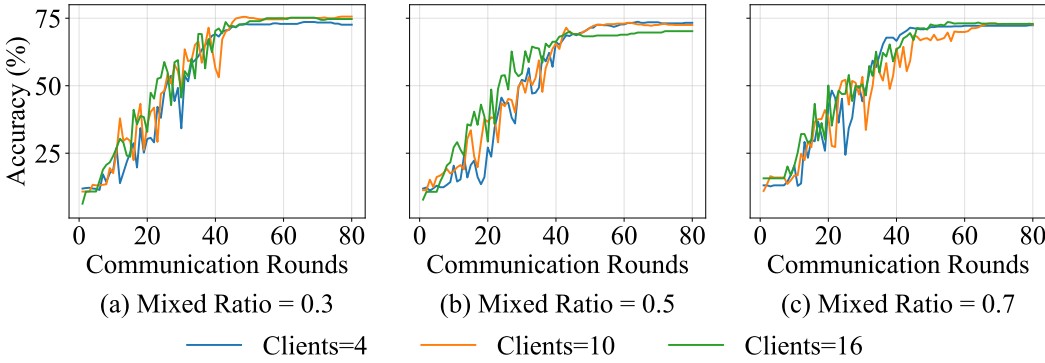

Figure 9: Convergence performance of FedPTE on Office with multi-domain.

## C.2 CONVERGENCE PERFORMANCE

Figure 9 demonstrates the convergence behavior of FedPTE in multi-domain mixed client scenarios, comparing performance curves under configurations of 4, 10, and 16 clients. FedPTE exhibits favorable convergence and stability across all settings. Specifically, when the communication rounds reach approximately 40, the model performance under all three client configurations stabilizes, with accuracy maintained at around 70%. Notably, even as the number of clients increases from 4 to 16, FedPTE maintains similar convergence speeds and final performance levels, indicating that its prototype topology evolution mechanism exhibits strong adaptability to system scale. During the early training phase (first 20 rounds), all configurations show rapid performance improvement, reflecting FedPTE's ability to effectively utilize initial communication rounds to quickly establish discriminative feature representations.

## C.3 DATA DISTRIBUTION

Figure 10 illustrates the data distribution across 5 clients in a FL scenario on Digit, which simultaneously faces challenges of data heterogeneity and domain shift. Here, each client belongs to a distinct

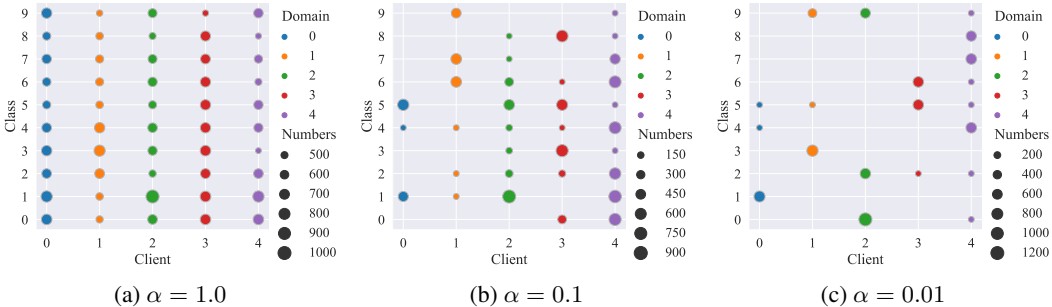

Figure 10: Examples of data distribution on Digit with 5 clients under different Dirichlet partitions $\alpha$.

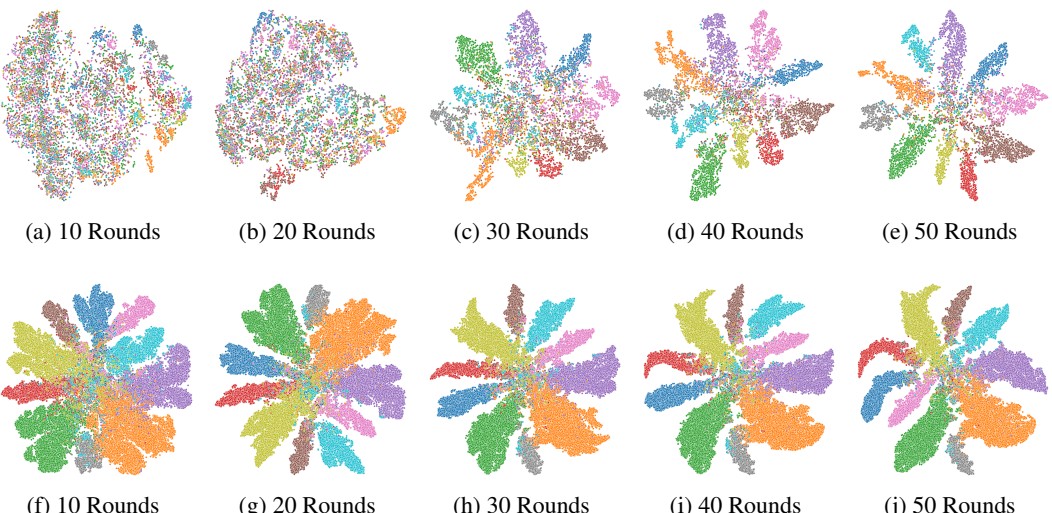

Figure 11: T-SNE Visualization of different communication rounds on Digit with 5 clients under data size ratio $= 0.1$ (a - e) and data size ratio $= 1.0$ (f - j).

domain, and the local data distribution follows a Dirichlet distribution controlled by the parameter $\alpha$. When $\alpha = 1.0$, the label distribution within each client is relatively uniform, approaching IID. In this case, the primary challenge for FL stems from the feature distribution differences caused by varying domains. In contrast, when $\alpha = 0.1$ or $\alpha = 0.01$, significant label heterogeneity arises among clients, where each client holds different categories and quantities of data. Under these conditions, FL must contend with the dual challenges of domain shift and label distribution skew.

## C.4 T-SNE VISUALIZATION

We explore the feature extraction performance of FedPTE across different sample ratios and communication rounds. Figure 11 shows the t-SNE (Maaten & Hinton, 2008) visualization of client data features. When clients are trained and tested on only 10% of Digit, the model fails to effectively separate features of different categories within the first 30 rounds. As training progresses to 30-50 rounds, the class boundaries in the feature space gradually become clearer, indicating that the model requires longer training cycles to learn effective representations with limited data.

In contrast, when trained with 100% of the data, the model achieves distinct in-class separation within just 10-20 rounds, with significantly improved compactness and discriminability of feature clusters. This demonstrates that sufficient data provides richer gradient signals, accelerating the formation of discriminative feature structures. In low-data scenarios, FedPTE ultimately achieves

stable feature alignment through progressive prototype topology evolution, highlighting its adaptability to data-scarce environments.

# D  METHOD ANALYSIS

## D.1  NUMERICAL COINCIDENCE

Semantic separability and variance structure changes stem from the continuous evolution of clients' local feature extractors during the federated training process. In the early stages of training, the model's feature representation capability is weak, and features of the same class may be entangled, forming a broad, high-variance unimodal distribution. As training progresses, the feature extractors are continuously optimized on their respective domain data, and their discriminative power gradually strengthens. This causes the previously entangled class features to begin developing clear substructures, manifesting as a refinement in semantic separability and an evolution in the variance structure. If prototypes remain static, they cannot capture this evolution. Global prototypes would become a blurred average of different sub-mode features, unable to provide precise anchors for any specific sub-mode. This leads to misalignment in feature alignment, where client models are pulled towards an inaccurate consensus point, thereby limiting further improvements in model performance.

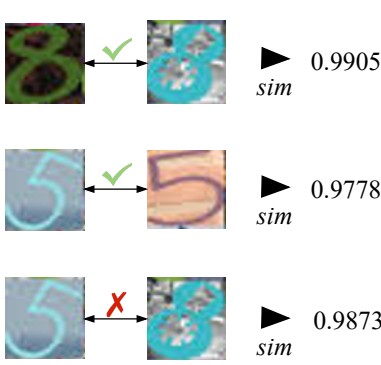

Figure 12: An example of the numerical coincidence.

The numerical coincidence refers to the scenario where two vectors are close in terms of Euclidean distance or cosine similarity in the feature space, yet their corresponding deep semantic information differs. This proximity is typically not due to genuine semantic similarity but is instead caused by irrelevant domain-specific features or training noise. As shown in Figure 12, in the feature space, the feature vector of a digit '5' might, due to a blue texture, unexpectedly lie close to the feature vector of a digit '8'. In such a case, if the prototype clusters representing '5' and '8' are merged based solely on spatial distance, different semantic categories will be confused, severely impairing the model's discriminative ability.

## D.2  COMMUNICATION OVERHEAD

The communication cost of FedPTE primarily consists of model parameter and prototype parameter transmission, which is similar to FPL and FedPLVM. Instead of uploading $M$ prototypes per client, each client uploads at most $M \times C_{\text{local}}$ cluster centers, each being a $d$-dimensional vector. Thus, the complexity of uploading prototypes is $O(M \cdot C_{\text{local}} \cdot d)$. Since ResNet has millions of parameters while the feature dimension $d = 512$ and $M = 10$, the condition $M \cdot C_{\text{local}} \cdot d \ll d_{\text{model}}$ holds, indicating that the additional communication overhead is relatively small and acceptable.

Table 13: Wall-clock results on Office with 4 clients.

| Methods | Wall-clock |
|---|---|
| FedOPT | 9.094s |
| FedDyn | 11.936s |
| Moon | 13.256s |
| FedProto | 12.154s |
| FPL | 15.973s |
| FedPLVM | 23.008s |
| FedHEAL | 18.547s |
| FedPall | 17.340s |
| FedPTE | 17.798s |

## D.3  COMPUTATIONAL COMPLEXITY

The main computational overhead on the server lies in topology evolution. For each prototype, clustering and marginal likelihood calculations are required. The marginal likelihood calculation involves the determinant and inverse of a covariance matrix, with a complexity of $O(d^3)$. The primary computational overhead on clients involves calculating the contrastive loss and performing clustering. The contrastive loss requires computing the similarity between samples and all global prototype cluster centers, with a complexity of $O(N_k \cdot |\mathcal{G}| \cdot d)$. We additionally test the

time required for FedPTE and baselines to complete local training and one communication round on a single A100 GPU, with results shown in Table 13. It can be observed that FedPTE maintains acceptable computational time while delivering expressive results.

Table 14: Comparison results on DomainNet with 6 clients. The best results are in bold.

| Methods | Clipart | Infograph | Painting | Quickdraw | Real | Sketch | Avg. |
|---|---|---|---|---|---|---|---|
| FedProto | 12.77 | 18.03 | 29.31 | 13.00 | 30.86 | 16.67 | 20.11 |
| FPL | 35.71 | 22.80 | 33.52 | 17.00 | 41.15 | 22.31 | 28.75 |
| FedPLVM | 55.19 | 33.16 | 45.05 | 22.67 | 52.70 | 42.56 | 41.89 |
| FedPTE | **58.85** | **33.44** | **53.25** | **31.40** | **59.22** | **46.19** | **47.06** |

### D.4 LARGE-SCALE PERFORMANCE

We compare the performance of prototype-based methods with FedPTE on the large-scale DomainNet (Peng et al., 2019), which comprises 6 domains, 345 classes per domain, and approximately 600000 images in total, as shown in Table 14. Specifically, all models undergo 200 communication rounds, with FedPTE's contrastive loss weight $\lambda$ set to 1. The results demonstrate that FedPTE achieves the best performance across all six domains, showcasing its significant advantages in handling complex cross-domain FL tasks. Traditional prototype-based methods rely on static clustering, making it difficult to capture potential multiple visual modes within a class. In contrast, FedPTE can adaptively split or merge prototype clusters based on statistical evidence, thereby more accurately modeling the multi-modal distributions within classes, which is particularly crucial in classification tasks with a large number of classes.

Table 15: Wall-clock results on DomainNet with 6 clients.

| Methods | Wall-clock |
|---|---|
| FedOPT | 514.436s |
| FedDyn | 499.409s |
| Moon | 731.948s |
| FedProto | 499.227s |
| FPL | 524.476s |
| FedPLVM | 808.268s |
| FedHEAL | 944.433s |
| FedPall | 745.655s |
| FedPTE | 726.983s |

Furthermore, on large-scale datasets, FedPTE's Bayesian evidence computation is parallelized. On a server with sufficient computational resources, the time required for topology evolution in each communication round depends not on the total number of prototype clusters, but on the time needed to process the single largest cluster. This enables FedPTE to efficiently handle complex semantic topologies containing a large number of prototype clusters. We supplement the computational times of FedPTE and the baselines on DomainNet in Table 15, showing that FedPTE's per-round computation time remains within a reasonable range.

## E THE USE OF LARGE LANGUAGE MODELS (LLMS)

During the preparation of this work, the authors use LLMs to improve the clarity and fluency of the writing. This assistance was limited to polishing grammar, rephrasing sentences, and enhancing readability.

