# OpenReview forum: "Bayesian Evidence-Driven Prototype Evolution for Federated Domain Adaptation"
_ICLR.cc/2026/Conference — ICLR 2026 Poster_

### Official Review · Reviewer_pRNn · 2025-10-26

**Soundness:** 3
**Presentation:** 2
**Contribution:** 2
**Rating:** 4
**Confidence:** 2

**Summary:**

This paper proposes FedPTE, a framework designed to address domain shift among clients. FedPTE dynamically adjusts the structure of global prototype clusters using Bayesian Gaussian Mixture Models (BGMM) and marginal likelihood ratios, determining when to split or merge prototypes based on statistical evidence. A stability constraint mechanism is introduced on the server to prevent unstable topology changes, while topology-aware contrastive learning on clients enhances feature discriminability and cross-domain consistency. Experiments demonstrate that FedPTE achieves generalization and stability.

**Strengths:**

1. The work targets a critical pain point in FL——domain shift under heterogeneous data, making the motivation both practical and research-relevant.
2. FedPTE maintains good performance under moderate privacy noise, suggesting potential for deployment in privacy-sensitive fields.

**Weaknesses:**

1. Relies on Gaussian mixture assumptions: performance may drop if data significantly deviates from this distribution.
2. Server-side Bayesian inference and covariance inversion have $O(d^3)$ complexity, which can be expensive in high-dimensional spaces.
3. Although probabilistic evidence is used, the paper lacks a deeper theoretical analysis of convergence guarantees or probabilistic bounds.
4. Lack of large-scale or real-world deployments where communication and noise issues are more severe.

**Questions:**

Please see the weaknesses.

---

> ### Author Response · Authors · 2025-11-20
> **Reply to Reviewer pRNn**
>
> Dear Reviewer pRNn,
>
> > **W1:** Relies on Gaussian mixture assumptions: performance may drop if data significantly deviates from this distribution.
>
> The core advantage of Gaussian Mixture Models lies in their ability to flexibly approximate various complex continuous distributions through multiple components. They are widely and successfully used to model complex, non-Gaussian real-world data. The goal is not to perfectly fit every data point, but to **capture the primary modes and covariance structure of the data distribution**. Additionally, the NIW prior possesses **heavy-tailed properties**, making it more robust to outliers and distribution skews that deviate from Gaussianity, thus avoiding the fragility that might arise from a single Gaussian model. As shown in Table 7 of the paper, FedPTE achieves significant performance improvements on multiple medical imaging datasets, demonstrating that the BGMM-based evolution mechanism is robust against the non-Gaussian distributions commonly encountered in practical applications.
>
> ||Kvasir|ETIS|ColonDB|ClinicDB|Avg.|
> |-|-|-|-|-|-|
> |FedOPT|51.62|58.21|55.81|59.25|56.22|
> |FedDyn|77.54|62.46|73.11|63.30|69.10|
> |Moon|49.87|56.35|58.94|54.63|54.95|
> |FedProto|78.25|63.40|71.93|66.63|70.05|
> |FPL|80.88|65.63|73.14|68.77|72.11|
> |FedPLVM|76.05|64.08|66.44|68.77|68.84|
> |FedHEAL|71.33|64.14|63.87|66.21|66.39|
> |FedPTE|**84.17**|**68.08**|**76.05**|**69.80**|**74.53**|
>
> *Table 1: Real-world domain results of performance comparison on Fed-Polyp with 4 clients.*
>
> ||BIDMC|BMC|HK|I2CVB|RUNMC|UCL|Avg.|
> |-|-|-|-|-|-|-|-|
> |FedOPT|50.04|50.31|50.90|50.86|50.53|50.43|50.51|
> |FedDyn|83.01|83.86|89.90|80.24|80.49|79.72|82.87|
> |Moon|85.82|78.51|86.16|81.44|78.34|77.31|81.26|
> |FedProto|85.18|88.48|87.40|81.65|86.60|85.31|85.77|
> |FPL|88.99|88.50|88.82|83.50|87.64|86.09|87.26|
> |FedPLVM|88.24|89.28|87.99|87.49|86.75|83.50|87.21|
> |FedHEAL|88.62|86.30|88.85|87.17|82.48|83.07|86.08|
> |FedPTE|**89.61**|**90.20**|**89.60**|**90.98**|**89.51**|**89.40**|**89.88**|
>
> *Table 2: Real-world domain results of performance comparison on Fed-Prostate with 6 clients.*
>
> > **W2:** Server-side Bayesian inference and covariance inversion have $O(d^3)$ complexity, which can be expensive in high-dimensional spaces.
>
> A feature dimension of $d=512$ is a typical setting in current FL feature alignment tasks. This is far smaller than the number of model parameters, and the number of prototypes $|\mathcal{G}|$ is usually not large. Therefore, the computational burden on the server is manageable. Meanwhile, the computational overhead of FedPTE is mainly **concentrated on the server, which typically possess more abundant computational resources than clients**. More importantly, topological evolution is not performed for every prototype cluster in every communication round. Operations are only triggered when there is sufficient statistical evidence and they pass the stability penalty checks, which effectively distributes the computational pressure per round. Furthermore, the Bayesian evidence computation is parallelized. On a server with sufficient computational resources, the time required for topology evolution in each communication round depends not on the total number of prototype clusters, but on the time needed to process the single largest cluster. This enables FedPTE to efficiently handle complex semantic topologies containing a large number of prototype clusters.
>
> > **W3:** Although probabilistic evidence is used, the paper lacks a deeper theoretical analysis of convergence guarantees or probabilistic bounds.
>
> Following your suggestion, we have supplemented the convergence proof in **Appendix A**.
>
> > **W4:** Lack of large-scale or real-world deployments where communication and noise issues are more severe.
>
> We compared the performance of FedPTE with baselines on real-world medical datasets in **Appendix B.2**. Images from different medical centers exhibit significant differences in acquisition devices, imaging protocols, and patient populations, introducing noise and distribution shifts that are far more complex and realistic than those in synthetic datasets. As can be seen from Figure 6 and Table 7, FedPTE's consistently leading performance on these datasets demonstrates its strong robustness to complex real-world data heterogeneity.
>
> Furthermore, we supplement its performance compared to baselines on the large-scale DomainNet dataset in **Appendix D.4**. The results, as shown in the table below, demonstrate that FedPTE achieves optimal performance.
>
> |Methods|Accuracy|
> |-|-|
> |FedProto|20.11|
> |FPL|28.75|
> |FedPLVM|41.89|
> |FedPTE|**47.06**|
>
> *Table 3: Comparison results on DomainNet with 6 clients.*

---

### Official Review · Reviewer_P4u6 · 2025-10-27

**Soundness:** 3
**Presentation:** 3
**Contribution:** 3
**Rating:** 6
**Confidence:** 3

**Summary:**

This paper addresses domain shift in federated learning by proposing FedPTE, a framework that dynamically evolves prototype topology through Bayesian inference. The key innovation is treating prototype clusters as variable topological units that can split or merge based on statistical evidence from Bayesian Gaussian Mixture Models with Normal-Inverse-Wishart priors. The framework includes penalty mechanisms to ensure stability and uses prototype topology-aware contrastive learning on clients to enhance feature alignment across domains.

**Strengths:**

1. The use of Bayesian inference with NIW priors for prototype topology evolution is mathematically principled and moves beyond simple averaging or static clustering approaches.
2. FedPTE consistently outperforms baselines, achieving 93.98% on Digit and 73.01% on Office, with particularly notable improvements on challenging domains.
3. The penalty terms for split/merge operations effectively balance adaptability with training stability, addressing a key challenge in dynamic prototype methods.
4.  The paper addresses important aspects like communication overhead, computational complexity, and privacy protection with differential privacy experiments.

**Weaknesses:**

1. The method introduces several hyperparameters (βsplit, βmerge, κ0, ν0, S0)
2.  The O(d³) complexity for marginal likelihood calculations could become prohibitive for high-dimensional features, though this isn't thoroughly discussed.
3. While the Bayesian framework is principled, the paper lacks convergence guarantees or theoretical analysis of when/why the topology evolution helps with domain shift.
4. Some of deign choices needs to be further clarified, e.g., Why use FINCH clustering initially? How sensitive is performance to the NIW prior initialization?

**Questions:**

Please find in the above weaknesses.

---

> ### Author Response · Authors · 2025-11-20
> **Reply to Reviewer P4u6**
>
> Dear Reviewer P4u6,
>
> > **W1:** The method introduces several hyperparameters (βsplit, βmerge, κ0, ν0, S0)
>
> The core of FedPTE is the calculation of the marginal likelihood ratio, which is a process of evidence comparison. As long as the NIW prior settings are not extremely incorrect, the marginal likelihoods under the two hypotheses change synchronously, making their ratio relatively robust to the prior's hyperparameters. We set the strength of the mean prior $\kappa_0$ to 0.1, indicating weak confidence in the initial prior mean and allowing the posterior distribution to be more easily dominated by the client prototype data uploaded in the current round. For the degrees of freedom $\nu_0$, according to the theoretical requirements of the NIW distribution, we set it to $\nu_0 = d + 5$, where $d$ is the feature dimension. This is one of the minimum values that satisfies the distribution definition and provides weak informativeness, again avoiding overly strong prior assumptions. For the scale matrix $\mathbf{S}_0$, we initialize it as $\mathbf{S}_0 = (\nu_0 - d - 1) \cdot I$, where $I$ is the identity matrix. This is a unit-scaled covariance prior, a common and robust default choice.
>
> Following your suggestion, we explore a strategy for adaptive penalty thresholds and have updated it in **Appendix B.4** of the paper. The core idea is to dynamically adjust the penalty strength based on intra-cluster compactness and inter-cluster separation. When the intra-cluster distance is relatively small, penalty constraints are appropriately relaxed to allow for more structural optimization. When the intra-cluster structure is loose, penalties are strengthened to avoid unstable topological changes. The results, as shown in the table below, indicate that the adaptive penalty mechanism demonstrates more adaptable performance in most cases. It automatically adjusts decision thresholds based on the actual characteristics of the cluster structure, reducing reliance on empirical hyperparameter tuning.
>
> ||MNIST|SVHN|USPS|Synth|MNIST-M|
> |-|-|-|-|-|-|
> |Adaptive|98.89|85.25|98.55|95.57|93.47|
> |Fixed|98.88|84.93|98.32|95.13|92.65|
>
> *Table 1: Adaptive weight results on Digit with 5 clients.*
>
> ||Amazon|Caltech|DSLR|Webcam|
> |-|-|-|-|-|
> |Adaptive|81.77|60.89|65.62|77.97|
> |Fixed|80.21|57.38|71.79|82.66|
>
> *Table 2: Adaptive weight results on Office with 4 clients.*
>
> > **W2:** The O(d³) complexity for marginal likelihood calculations could become prohibitive for high-dimensional features, though this isn't thoroughly discussed.
>
> A feature dimension of $d=512$ is a typical setting in current FL feature alignment tasks. This is far smaller than the number of model parameters, and the number of prototypes $|\mathcal{G}|$ is usually not large. Therefore, the computational burden on the server is manageable. Meanwhile, the computational overhead of FedPTE is mainly concentrated on the server side. In the FL setting, **servers typically possess more abundant computational resources than clients**. More importantly, topological evolution is not performed for every prototype cluster in every communication round. Operations are only triggered when there is sufficient statistical evidence and they pass the stability penalty checks, which effectively distributes the computational pressure per round. Furthermore, the Bayesian evidence computation is parallelized. On a server with sufficient computational resources, the time required for topology evolution in each communication round depends not on the total number of prototype clusters, but on the time needed to process the single largest cluster. This enables FedPTE to efficiently handle complex semantic topologies containing a large number of prototype clusters.
>
> > **W3:** While the Bayesian framework is principled, the paper lacks convergence guarantees or theoretical analysis of when/why the topology evolution helps with domain shift.
>
> Following your suggestion, we have supplemented the convergence proof in **Appendix A**.

---

> > ### Author Response · Authors · 2025-11-20
> >
> > > **W4:** Some of deign choices needs to be further clarified, e.g., Why use FINCH clustering initially? How sensitive is performance to the NIW prior initialization?
> >
> > We chose the FINCH on clients primarily since it does not require predefining the number of clusters, thus avoiding the unreasonableness of manually setting cluster values. This provides an adaptive starting point for the subsequent Bayesian topology evolution on the server. FINCH's complexity is nearly linear, making it suitable for running on clients with limited computational resources. This matches the typical computational environment of FL clients. Therefore, using FINCH allows extracting the most representative, fine-grained local data structures from each client without introducing additional hyperparameters or excessive computational burden.
> >
> > In **Appendix B.3**, we explore the impact of different prior hyperparameter values on model performance, with results shown in the tables below. The results indicate that as long as these parameters are set within reasonable ranges, maintaining the nature of weakly informative priors, their variation has minimal impact on the final topology evolution decisions and model performance.
> >
> > ||MNIST|SVHN|USPS|Synth|MNIST-M|Avg.|
> > |-|-|-|-|-|-|-|
> > |$\kappa_0=0.01$|98.94|84.41|98.28|94.97|91.92|93.71|
> > |$\kappa_0=0.05$|98.94|84.46|98.49|95.11|91.43|93.69|
> > |$\kappa_0=0.1$|98.88|84.93|98.32|95.13|92.65|93.98|
> > |$\kappa_0=0.5$|98.96|84.52|98.44|95.22|91.86|93.80|
> > |$\kappa_0=1.0$|98.96|84.65|98.28|95.05|92.25|93.84|
> >
> > *Table 3: NIW Prior hyper-parameter $\kappa_0$ ablation results on Digit with 5 clients.*
> >
> > ||MNIST|SVHN|USPS|Synth|MNIST-M|Avg.|
> > |-|-|-|-|-|-|-|
> > |$\nu_0=d+2$|98.93|83.93|98.33|95.04|91.94|93.63|
> > |$\nu_0=d+5$|98.88|84.93|98.32|95.13|92.65|93.98|
> > |$\nu_0=d+10$|99.04|84.66|98.55|95.12|92.10|93.89|
> > |$\nu_0=d+20$|98.91|84.20|98.55|95.12|92.16|93.79|
> >
> > *Table 4: NIW Prior hyper-parameter $\nu_0$ ablation results on Digit with 5 clients.*

---

### Official Review · Reviewer_y5F7 · 2025-10-29

**Soundness:** 3
**Presentation:** 2
**Contribution:** 2
**Rating:** 4
**Confidence:** 4

**Summary:**

FedPTE frames federated domain adaptation as evidence-driven prototype topology evolution. The server uses BGMM with NIW prior and marginal likelihood ratio to split/merge global prototypes, adding a stability penalty; the client uses topology-aware contrastive loss for training. This has achieved performance improvements on other feature shift datasets such as Digit/Office.

**Strengths:**

1. The article mainly focuses on the fact that static clustering methods are difficult to solve the semantic separability and variance structure changes between prototype clusters in feature offset scenarios, and from the experimental results, it has excellent performance, and the experiment is complete.
2. The ablation experiment verified the effectiveness of the effects of each component, which solved my confusion to a certain extent.

**Weaknesses:**

1. The causes and impacts of semantic separability and variance structure changes under feature transfer have not been fully explained. In addition, how should I understand the concept of “numerical coincidence” mentioned in Section 3.2? Why does this lead to situations where spatial proximity but different semantics occur?
2. The ablation experiments were not performed on the same experimental dataset, which makes me speculate about the method's effectiveness. For example, based on Table 3 and Figure 4, I cannot judge the effect of the stability component without adding $P_{split}$ and $P_{merge}$ and adjusting their corresponding parameters.
3. In Formula 2, the calculation of local prototypes is done by generating different feature centers through FINCH clustering, while the left side is the local prototypes of samples of the same type, which are not equal in number.
4. Though the authors have extended experiments on mixed data from more clients in the appendix, the main results are conducted in a small-scale environment (5 clients), limiting the reliability of the results and raising potential scalability concerns of the proposed method.
5. Comparison with some latest SOTA methods on federated domain shift is not included, for example:
[1]  "FedPall: Prototype-based Adversarial and Collaborative Learning for Federated Learning with Feature Drift." ICCV, 2025.
6. The distinction between related work and other work should be further clarified.

**Questions:**

1. More ablation experiments are needed to verify whether the reference of the stability constraint mechanism will affect the specific performance.
2. What specific weaknesses are addressed relative to FedPLVM/MPFT? And what are the advantages of NIW proofs compared to previous heuristics?
3. "Numerical coincidence": Give an example where spatial proximity hides a semantic mismatch or provide insight into why this happens and how to prevent it through penalties.
4. Reports communication and runtime comparisons to prototype baselines.

---

> ### Author Response · Authors · 2025-11-20
> **Reply to Reviewer y5F7**
>
> Dear Reviewer y5F7,
>
> > **W1 (C3):** The causes and impacts of semantic separability and variance structure changes under feature transfer have not been fully explained. In addition, how should I understand the concept of “numerical coincidence” mentioned in Section 3.2? Why does this lead to situations where spatial proximity but different semantics occur?
>
> Semantic separability and variance structure changes stem from the continuous evolution of clients' local feature extractors during the federated training process. In the early stages of training, the model's feature representation capability is weak, and features of the same class may be entangled, forming a broad, high-variance unimodal distribution. As training progresses, the feature extractors are continuously optimized on their respective domain data, and their discriminative power gradually strengthens. This causes the previously entangled class features to begin developing clear substructures, manifesting as a refinement in semantic separability and an evolution in the variance structure. If prototypes remain static, they cannot capture this evolution. Global prototypes would become a blurred average of different sub-mode features, unable to provide precise anchors for any specific sub-mode. This leads to misalignment in feature alignment, where client models are pulled towards an inaccurate consensus point, thereby limiting further improvements in model performance.
>
> Numerical coincidence refers to the scenario where two vectors are close in terms of Euclidean distance or cosine similarity in the feature space, yet their corresponding deep semantic information differs. This proximity is typically not due to genuine semantic similarity but is instead caused by irrelevant domain-specific features or training noise. Following your suggestion, we have added an explanation in **Appendix D.1**. As illustrated in the feature space, the feature vector of a digit '5' might, due to a blue texture, unexpectedly lie close to the feature vector of a digit '8'. In such a case, if the prototype clusters representing '5' and '8' are merged based solely on spatial distance, different semantic classes will be confused, severely impairing the model's discriminative ability. FedPTE introduces a semantic penalty term based on KL divergence to check the distributional consistency between two clusters, effectively distinguishing between cases of accidental spatial proximity and genuine semantic similarity, thereby preventing the erroneous merging of clusters that are semantically different but numerically close.
>
> > **W2 (C1):** The ablation experiments were not performed on the same experimental dataset, which makes me speculate about the method's effectiveness. For example, based on Table 3 and Figure 4, I cannot judge the effect of the stability component without adding $P_{split}$ and $P_{merge}$ and adjusting their corresponding parameters.
>
> We have supplemented the results for the penalty threshold hyperparameters on the Digit dataset in **Appendix B.4**, as shown in the table below. FedPTE exhibits a certain degree of robustness to the selection of penalty coefficients, maintaining relatively stable performance within the ranges of $\beta_{\text{split}} \in [0.5, 1.5]$ and $\beta_{\text{merge}} \in [1.0, 2.0]$.
>
> ||$\beta_{\text{split}}=0.5$|$\beta_{\text{split}}=1.0$|$\beta_{\text{split}}=1.5$|$\beta_{\text{split}}=2.0$|
> |-|-|-|-|-|
> |$\beta_{\text{merge}}=2.0$|92.53|93.52|93.29|92.01|
> |$\beta_{\text{merge}}=1.5$|93.71|93.98|93.64|92.72|
> |$\beta_{\text{merge}}=1.0$|92.68|92.90|93.33|92.83|
> |$\beta_{\text{merge}}=0.5$|91.79|92.65|92.85|92.35|
>
> *Table 1: Penalty ablation results on Digit with 5 clients.*
>
> > **W3:** In Formula 2, the calculation of local prototypes is done by generating different feature centers through FINCH clustering, while the left side is the local prototypes of samples of the same type, which are not equal in number.
>
> Equation (2) describes not the process of generating the entire set $\mathcal{P}^m_k$, but the method for calculating each individual prototype vector $\mathbf{p}^{m,c}_k$ within that set, obtained via FINCH. To avoid ambiguity, we have modified $\mathbf{p}^m_k$ to $\mathbf{p}^{m,c}_k$ in the methodology section, where the superscripts $m$ and $c$ together specify that the current calculation is for the prototype of the $c$-th cluster under class $m$.

---

> > ### Author Response · Authors · 2025-11-20
> >
> > > **W4:** The main results are conducted in a small-scale environment (5 clients), limiting the reliability of the results and raising potential scalability concerns of the proposed method.
> >
> > We have supplemented the multi-client experiments in **Appendix B.1**, including results under both domain-balanced and domain-imbalanced settings, as shown in the tables below. As the number of clients increases, FedPTE's accuracy slightly decreases but consistently and significantly outperforms all baseline methods.
> >
> > ||5 Clients|10 Clients|15 Clients|20 Clients|25 Clients|
> > |-|-|-|-|-|-|
> > |FedProto|87.52|78.48|76.71|68.64|62.43|
> > |FPL|90.10|82.03|77.14|75.60|72.03|
> > |FedPLVM|91.55|90.34|89.15|87.86|85.95|
> > |FedPTE|**93.98**|**93.75**|**93.12**|**92.87**|**91.92**|
> >
> > *Table 2: Domain-balanced results on Digit with various numbers of clients.*
> >
> > ||10 Clients|15 Clients|20 Clients|25 Clients|
> > |-|-|-|-|-|
> > |FedProto|74.03|70.23|59.62|52.70|
> > |FPL|75.85|74.87|65.77|64.84|
> > |FedPLVM|86.18|85.52|82.67|79.37|
> > |FedPTE|**88.66**|**88.46**|**86.97**|**84.37**|
> >
> > *Table 3: Domain-imbalanced results on Digit with various numbers of clients.*
> >
> > > **W5:** Comparison with some latest SOTA methods on federated domain shift is not included.
> >
> > The official publication date of the paper was after the ICLR paper submission deadline. Following your suggestion, we have added a comparison with FedPall in **Section 4.2**, as shown in the table below. It can be observed that FedPTE still maintains the best performance.
> >
> > ||MNIST|SVHN|USPS|Synth|MNIST-M|Avg.|
> > |-|-|-|-|-|-|-|
> > |FedPall|97.99±0.38|73.70±1.21|91.88±0.70|96.37±0.19|77.48±0.05|87.48|
> > |FedPTE|**98.88±0.07**|**84.93±0.17**|**98.32±0.07**|**95.13±0.07**|**92.65±0.09**|**93.98**|
> >
> > *Table 4: Main results of performance comparison on Digit with 5 clients.*
> >
> > ||Amazon|Caltech|DSLR|Webcam|Avg.|
> > |-|-|-|-|-|-|
> > |FedPall|76.21±1.59|51.41±4.20|66.67±1.86|67.82±1.99|65.53|
> > |FedPTE|**80.21±0.43**|**57.38±0.91**|**71.79±3.90**|**82.66±1.38**|**73.01**|
> >
> > *Table 5: Main results of performance comparison on Office with 4 clients.*
> >
> > > **W6 (C2):** The distinction between related work and other work should be further clarified.
> >
> > FedPLVM performs clustering independently in each round, lacking continuity across rounds. The prototype structure fluctuates drastically between rounds, preventing gradual semantic refinement. FedPTE introduces a continuous prototype topology evolution mechanism that can intelligently decide when to split, when to merge, and when to remain unchanged based on Bayesian evidence, thereby stably adapting to the dynamic changes in the feature space. MPFT relies on the performance of pre-trained models and requires explicit domain identity information to generate domain-specific prototypes. This limits its applicability in real-world scenarios that lack high-quality pre-trained models or have blurred domain boundaries. FedPTE is data-driven and domain-agnostic, not relying on any pre-trained models. It automatically discovers the intrinsic semantic structure of the data through statistical evidence, thus exhibiting better applicability in a wider range of heterogeneous scenarios.
> >
> > Traditional heuristic methods rely on empirical hyperparameters, and their decision-making process lacks theoretical grounding. We have supplemented an ablation study on the effectiveness of the NIW prior in **Appendix B.3**, with results shown in the table below. Specifically, we replace it with random evolution and a simple heuristic to compare against FedPTE's performance. The results show that using the NIW prior achieves the best performance, outperforming the other variants across all domains.
> >
> > ||Amazon|Caltech|DSLR|Webcam|Avg.|
> > |-|-|-|-|-|-|
> > |Random|67.19|53.78|46.88|54.24|55.52|
> > |Simple|75.00|54.22|56.25|66.10|62.89|
> > |NIW|**80.21**|**57.38**|**71.79**|**82.66**|**73.01**|
> >
> > *Table 6: Prior component ablation results on Office with 4 clients.*
> >
> > > **C4:** Reports communication and runtime comparisons to prototype baselines.
> >
> > We have already analyzed the communication overhead and compared the computation time for FedPTE in **Appendices D.2 and D.3**, respectively. Since the number of prototypes uploaded per communication round by prototype-based methods and FedPTE is not fixed, we cannot provide specific numerical values for communication volume. It is worth noting that the parameter size of prototypes is **far smaller than that of the model parameters**, so it can be considered that the communication overhead required by prototype-based methods and FedPTE is similar. According to the computation time results, FedPTE achieves superior performance while maintaining computational overhead comparable to the baselines.
> >
> > |Methods|Wall-clock|Accuracy|
> > |-|-|-|
> > |FedOPT|9.094s|27.90|
> > |FedDyn|11.936s|53.97|
> > |Moon|13.256s|58.64|
> > |FedProto|12.154s|56.29|
> > |FPL|15.973s|52.04|
> > |FedPLVM|23.008s|67.86|
> > |FedHEAL|18.547s|58.87|
> > |FedPall|17.340s|65.53|
> > |FedPTE|17.798s|73.01|
> >
> > *Table 7: Comparison results on Office with 4 clients.*

---

### Official Review · Reviewer_Pn6w · 2025-11-02

**Soundness:** 3
**Presentation:** 3
**Contribution:** 3
**Rating:** 6
**Confidence:** 3

**Summary:**

This paper introduces FedPTE, a Bayesian prototype-topology–based federated learning (FL) framework designed to tackle domain shift and data heterogeneity in cross-domain FL. Traditional prototype-based FL methods rely on static clustering or averaging and fail to adapt to evolving class structures.
FedPTE models prototypes as dynamic topological units whose number and structure evolve via Bayesian Gaussian Mixture Models (BGMM) and marginal likelihood ratio tests under a Normal–Inverse–Wishart (NIW) prior. The server decides whether to split or merge prototype clusters based on evidence strength, guided by penalty constraints to maintain stability. Clients perform prototype topology–aware contrastive learning, aligning local representations with global prototypes.

Experiments on Digit (MNIST/SVHN/USPS/Synth/MNIST-M), Office (Amazon/Caltech/DSLR/Webcam), and multiple medical imaging datasets (Camelyon, ISIC, Polyp, Prostate) show that FedPTE outperforms baselines such as FedProto, FedPLVM, FedHEAL, and FPL. The paper also reports ablation studies, non-IID analysis, privacy perturbation effects, and multi-domain scalability. Overall, FedPTE claims better adaptability, robustness, and generalization across heterogeneous domains.

**Strengths:**

1 Original idea: Introduces Bayesian evidence to dynamically evolve prototype topology—bridging probabilistic inference with federated representation learning.

2 Comprehensive experiments: Covers IID/Non-IID, unseen domain, medical imaging, and privacy scenarios.

3 Strong empirical performance: Consistent gains over FedProto, FedPLVM, FPL across all datasets.

4 Well-designed ablations: Component and hyperparameter studies are clear (Figure 3–4, Table 3).

**Weaknesses:**

1 Limited novelty beyond existing prototype frameworks.
While Bayesian modeling adds rigor, the overall structure (prototype aggregation + contrastive alignment) resembles prior works like FedPLVM and FPL. The innovation lies mainly in the split/merge inference.

2 Assumption of Gaussian prototypes.
Real FL data distributions (e.g., in medical or digit datasets) may be non-Gaussian; the NIW assumption can oversimplify multi-modal or skewed clusters.

3 Hyperparameter sensitivity.
Penalty coefficients (β_split, β_merge) and evidence thresholds are empirically set; no adaptive rule or theoretical guidance is provided.

4 Computational overhead.
Marginal likelihood evaluation (O(d³)) and frequent topology updates could become expensive for large models or high-dimensional features, though claimed “acceptable” (Table 10) without quantitative scaling analysis.

5 No theoretical convergence guarantee.
The stability of prototype evolution across rounds is discussed qualitatively but lacks formal proof or bound.

**Questions:**

1 How sensitive is FedPTE to NIW hyperparameters (κ₀, ν₀, S₀)? Are priors fixed or updated adaptively across rounds?

2 Could the split/merge evidence thresholds be learned dynamically rather than hard-coded?

3 Does the BGMM assumption hold when feature distributions are highly non-Gaussian (e.g., with ReLU activation skew)?

4 How does FedPTE behave when one domain is highly dominant (client imbalance)?

5 Is the Bayesian evidence computation parallelized, and what is its runtime complexity per round in large-scale FL?

---

> ### Author Response · Authors · 2025-11-20
> **Reply to Reviewer Pn6w**
>
> Dear Reviewer Pn6w,
>
> > **W1:** Limited novelty beyond existing prototype frameworks.
>
> Existing methods such as FedPLVM and FPL rely on static clustering, making it difficult to capture the dynamic evolution of intra-class structures during training. FedPTE is the first to treat prototypes as evolvable topological units. It employs Bayesian Gaussian Mixture Models and marginal likelihood ratio tests under an NIW prior for statistical inference, deciding whether to split or merge prototype clusters, thereby enabling finer-grained modeling of cross-domain feature distributions. Furthermore, FedPTE introduces penalty terms for splitting and merging to balance the flexibility of topological evolution with training stability, effectively preventing semantic inconsistency issues caused by numerical coincidence.
>
> > **W2 (C3):** Assumption of Gaussian prototypes.
>
> The core advantage of Gaussian Mixture Models lies in their ability to flexibly approximate various complex continuous distributions through multiple components. They are successfully used to model complex, non-Gaussian real-world data. The goal is not to perfectly fit every data point, but to **capture the primary modes and covariance structure of the data distribution**. Additionally, the NIW prior possesses **heavy-tailed properties**, making it more robust to outliers and distribution skews that deviate from Gaussianity, thus avoiding the fragility that might arise from a single Gaussian model. As shown in Table 1 and Table 7 of the paper, FedPTE achieves significant performance improvements on Digit and multiple medical imaging datasets, demonstrating that the BGMM-based evolution mechanism is robust against the non-Gaussian distributions commonly encountered in practical applications.
>
> > **W3 (C2):** Hyperparameter sensitivity.
>
> Following your suggestion, we explore a strategy for adaptive penalty thresholds and have updated it in **Appendix B.4**. The core idea is to dynamically adjust the penalty strength based on intra-cluster compactness and inter-cluster separation. When the intra-cluster distance is relatively small, penalty constraints are appropriately relaxed to allow for more structural optimization. When the intra-cluster structure is loose, penalties are strengthened to avoid unstable topological changes. The results, as shown in the table below, indicate that the adaptive penalty mechanism demonstrates more adaptable performance in most cases. It automatically adjusts decision thresholds based on the actual characteristics of the cluster structure, reducing reliance on empirical hyperparameter tuning.
>
> ||MNIST|SVHN|USPS|Synth|MNIST-M|
> |-|-|-|-|-|-|
> |Adaptive|98.89|85.25|98.55|95.57|93.47|
> |Fixed|98.88|84.93|98.32|95.13|92.65|
>
> *Table 1: Adaptive weight results on Digit with 5 clients.*
>
> ||Amazon|Caltech|DSLR|Webcam|
> |-|-|-|-|-|
> |Adaptive|81.77|60.89|65.62|77.97|
> |Fixed|80.21|57.38|71.79|82.66|
>
> *Table 2: Adaptive weight results on Office with 4 clients.*
>
> > **W4 (C5):** Computational overhead.
>
> In our experiments, the feature dimension $d$ is set to 512. For $d=512$, the inversion and determinant calculation of the covariance matrix are highly optimized on modern computing hardware. Meanwhile, the computational overhead of FedPTE is mainly **concentrated on the server, which typically possess more abundant computational resources than clients**. More importantly, topological evolution is not performed for every prototype cluster in every communication round. Operations are only triggered when there is sufficient statistical evidence and they pass the stability penalty checks, which effectively distributes the computational pressure per round.
>
> As shown in Table 13 of the paper, the time for FedPTE (17.798s) falls between FPL (15.973s) and FedPLVM (23.008s), which is within a reasonable range. Considering the performance gains brought by FedPTE, we believe the limited additional computational overhead is cost-effective. Furthermore, the Bayesian evidence computation is **parallelized**. On a server with sufficient computational resources, the time required for topology evolution in each communication round depends not on the total number of prototype clusters, but on the time needed to process the single largest cluster. This enables FedPTE to efficiently handle complex semantic topologies containing a large number of prototype clusters. Regarding your concern about FedPTE's computational overhead in large-scale FL, we have supplemented in **Appendix D.4** its computational time and performance compared to baselines on the large-scale DomainNet dataset. The results, as shown in the table below, demonstrate that FedPTE achieves optimal performance within a reasonable computation time frame.
>
> |Methods|Wall-clock|Accuracy|
> |-|-|-|
> |FedProto|499.227s|20.11|
> |FPL|524.476s|28.75|
> |FedPLVM|808.268s|41.89|
> |FedPTE|726.983s|47.06|
>
> *Table 3: Comparison results on DomainNet with 6 clients.*

---

> > ### Author Response · Authors · 2025-11-20
> >
> > > **W5:** No theoretical convergence guarantee.
> >
> > Following your suggestion, we have supplemented the convergence proof in **Appendix A**.
> >
> > > **C1:** How sensitive is FedPTE to NIW hyperparameters (κ₀, ν₀, S₀)? Are priors fixed or updated adaptively across rounds?
> >
> > FedPTE employs fixed priors to make computations independent of each other and to avoid maintaining complex states across rounds. Specifically, we set the strength of the mean prior $\kappa_0$ to 0.1, indicating weak confidence in the initial prior mean and allowing the posterior distribution to be more easily dominated by the client prototype data uploaded in the current round. For the degrees of freedom $\nu_0$, according to the theoretical requirements of the NIW distribution, we set it to $\nu_0 = d + 5$, where $d$ is the feature dimension. This is one of the minimum values that satisfies the distribution definition and provides weak informativeness, again avoiding overly strong prior assumptions. For the scale matrix $\mathbf{S}_0$, we initialize it as $\mathbf{S}_0 = (\nu_0 - d - 1) \cdot I$, where $I$ is the identity matrix. This is a unit-scaled covariance prior, a common and robust default choice.
> >
> > In **Appendix B.3**, we explore the impact of different prior hyperparameter values on model performance, with results shown in the table below. The results indicate that as long as these parameters are set within reasonable ranges, maintaining the nature of weakly informative priors, their variation has minimal impact on the final topology evolution decisions and model performance.
> >
> > ||MNIST|SVHN|USPS|Synth|MNIST-M|Avg.|
> > |-|-|-|-|-|-|-|
> > |$\kappa_0=0.01$|98.94|84.41|98.28|94.97|91.92|93.71|
> > |$\kappa_0=0.05$|98.94|84.46|98.49|95.11|91.43|93.69|
> > |$\kappa_0=0.1$|98.88|84.93|98.32|95.13|92.65|93.98|
> > |$\kappa_0=0.5$|98.96|84.52|98.44|95.22|91.86|93.80|
> > |$\kappa_0=1.0$|98.96|84.65|98.28|95.05|92.25|93.84|
> >
> > *Table 4: NIW Prior hyper-parameter $\kappa_0$ ablation results on Digit with 5 clients.*
> >
> > ||MNIST|SVHN|USPS|Synth|MNIST-M|Avg.|
> > |-|-|-|-|-|-|-|
> > |$\nu_0=d+2$|98.93|83.93|98.33|95.04|91.94|93.63|
> > |$\nu_0=d+5$|98.88|84.93|98.32|95.13|92.65|93.98|
> > |$\nu_0=d+10$|99.04|84.66|98.55|95.12|92.10|93.89|
> > |$\nu_0=d+20$|98.91|84.20|98.55|95.12|92.16|93.79|
> >
> > *Table 5: NIW Prior hyper-parameter $\nu_0$ ablation results on Digit with 5 clients.*
> >
> > > **C4:** How does FedPTE behave when one domain is highly dominant (client imbalance)?
> >
> > We have supplemented the results for domain imbalance in **Appendix B.1**, as shown in the table below. Specifically, we configure at least half of the clients to be in the same domain, with the remaining clients proportionally distributed across the other domains. When at least half of the clients reside in the same domain, the data distribution exhibits skewness. FedPTE's performance decline is relatively small under these conditions, and it remains significantly superior to other methods. This demonstrates FedPTE's strong adaptability under skewed data distributions.
> >
> > ||10 Clients|15 Clients|20 Clients|25 Clients|
> > |-|-|-|-|-|
> > |FedProto|74.03|70.23|59.62|52.70|
> > |FPL|75.85|74.87|65.77|64.84|
> > |FedPLVM|86.18|85.52|82.67|79.37|
> > |FedPTE|**88.66**|**88.46**|**86.97**|**84.37**|
> >
> > *Table 6: Domain-imbalanced results on Digit with various numbers of clients.*

---

### Author Response · Authors · 2025-11-26
**Summary of Revisions**

Dear Reviewers,

We sincerely thank you for your thoughtful and constructive feedback on our submission. We have carefully considered all of your comments and have substantially revised the manuscript to address the concerns raised.

In the main body of the paper, we modified the notation of prototypes in **Section 3 (Preliminaries)** to eliminate ambiguity and added comparisons with the latest baseline FedPall in **Section 4.2** to demonstrate the effectiveness of FedPTE. Additionally, we relocated the "Multi-domain Results" and "Unseen Domain Results" from the appendix to **Sections 4.4 and 4.5**, respectively, to better illustrate the robustness of FedPTE under various domain shift scenarios.

In the appendices, we incorporated a convergence analysis in **Appendix A** to strengthen our theoretical contributions. We also introduced experiments in **Appendix B.1** to evaluate FedPTE's performance under varying numbers of clients, including both domain-balanced and domain-imbalanced settings. In **Appendix B.3**, we explored the sensitivity of NIW prior hyper-parameters and conducted ablation studies on the NIW prior component, demonstrating its robustness to hyper-parameter variations and its superiority over simple heuristics. In **Appendix B.4**, we enhanced the experiments on penalty coefficients and investigated an adaptive penalty strategy to reduce reliance on empirical hyper-parameter tuning. In **Appendix D.1**, we provided an in-depth explanation of numerical coincidence supplemented with a visual example. Finally, in **Appendix D.4**, we expanded the evaluation to include the large-scale DomainNet dataset and provided wall-clock time comparisons between FedPTE and baselines to further support the computational overhead analysis of FedPTE.

We believe the revised manuscript has been strengthened by your valuable suggestions, and we hope our revisions have satisfactorily addressed your concerns. Your insights are invaluable to us, and we remain fully committed to addressing any remaining points to further improve our work.

---

### Meta-Review · Area_Chair_qDar · 2026-01-04

**Summary:**

FedPTE is a federated learning framework that addresses domain shift by dynamically evolving prototype clusters using Bayesian evidence and marginal likelihood ratios for adaptive topology adjustments. It incorporates stability constraints on the server to prevent abrupt changes and uses prototype-aware contrastive learning on clients to enhance feature discriminability and cross-domain consistency. Experimental results show superior performance across various heterogeneous datasets, demonstrating its strong generalization and expressiveness in federated domain adaptation.

The primary concerns raised by reviewers pertain to the algorithm's convergence guarantees, parameter sensitivity analysis, assumptions about data distribution, and adaptability to large-scale clients and datasets. The authors have provided thorough explanations addressing these issues, including convergence analysis, additional parameter sensitivity experiments, validation of performance on non-Gaussian distributions, and performance testing on large-scale clients and datasets. The authors' rebuttal is thorough and effectively addresses the reviewers' concerns. I support its acceptance.

**Reviewer Concerns:**

The authors provided detailed and satisfactory responses to reviewer Pn6w's concerns regarding innovation, convergence, and parameter sensitivity analysis.

The authors offered good responses to reviewer y5F7's questions about comparative advantages, communication overhead, and operational efficiency. However, their reply to the issue of numerical coincidence lacked specificity.

Regarding reviewer P4u6's concerns about parameter analysis and clustering method selection, the authors' responses are relatively targeted, but they do not delve deeply into how high-dimensional features affect the algorithm.

Regarding reviewer pRNn's questions about distribution assumptions and large-scale validation, the authors provide thorough responses.

Overall, the authors' responses are detailed and well-targeted.

**Reviewer Scores:**

None of the reviewers added new comments to the authors' responses. However, in my view, the authors have provided thorough and satisfactory replies to the issues raised by reviewer pRNn, who is likely to increase their score. As for reviewer y5F7, since the authors have partially addressed their concerns, this reviewer may maintain their current score. Overall, the authors' responses are persuasive, and I support the acceptance of this paper.

---

### Decision · Program_Chairs · 2026-01-26

Accept (Poster)